



# Switch of fungal to bacterial degradation in natural, drained and rewetted oligotrophic peatlands reflected in d15N and fatty acid composition

Miriam Groß-Schmölders[1], Pascal von Sengbusch[2], Jan Paul Krüger[3], Kristy Woodard[4], Axel Birkholz[1], Jens Leifeld[4], Christine Alewell[1]

[1]Department of Environmental Geoscience, University of Basel, Basel, 4056, Switzerland
[2]Office for ecological reports, Kandern, 79400, Germany
[3]UDATA GmbH – Environment & Education, Neustadt a. d. Weinstraße, 67433, Germany
[4]Agroscope, Zürich, 8046, Switzerland

*Correspondence to*: Miriam Groß-Schmölders (miriam.gross-schmoelders@unibas.ch)

**Abstract.** During the last centuries major parts of European peatlands were degraded along with drainage and land use changes. Peatland biodiversity and essential ecosystem functions (e.g. flood prevention, groundwater purification and CO2 sink) were dramatically impaired. Moreover, climate change threatens peatlands in the near future. Increasing pressure to
peatland ecosystems calls for a more cost-efficient method to indicate the current state of peatlands and the success of restoration effort. Metabolism processes in peatland soils are imprinted in stable isotope signatures due to differences in microorganism communities and their metabolic pathways. Therefore we hypothesize that depth profiles of nitrogen stable isotope values provide a promising opportunity to detect peatland decomposition or restoration. We studied five peatlands: Degerö Stormyr (Northern Sweden), Lakkasuo (Central Finland) and three mires in the Black Forest (Southern Germany).
At all locations cores were taken from adjacent drained (or rewetted) and natural sites to identify $\partial 15N$ trends that could indicate changes due to drainage and restoration. At all drained (and rewetted) sites we found a distinct peak ("turning point") of the $\partial 15N$ values in the center of the drained horizon. To verify our interpretation $\partial 13C$, the C/N ratio and the bulk density were measured and a microscopic analysis of the macro residuals in the peat cores was made. In addition we did a phospholipid fatty acid (PLFAs) analysis to link our results to microbial community composition. We distinguished between
fungal and bacterial-derived PLFAs. In accordance with other studies, our results suggest, that fungi dominate the microbial metabolism in the upper, aerobic peat horizon. This is reflected by depleted $\partial 15N$ values. Downwards the drained horizon conditions slowly switch to oxygen limitation. In consequence fungal-derived PLFAs decreases whereas bacterial-derived PLFAs are rising. The highest diversity of microbial-derived PLFAs is indicated by the $\partial 15N$ turning point. Below the $\partial 15N$ turning point, oxygen is increasingly limited and concentrations of all microbial-derived PLFAs are decreasing down to the
onset of the permanently waterlogged, anaerobic horizon. Peatland cores with restoration success show, above the formerly



drainage-affected horizon, again no depth trend of the isotopic values. Hence, we conclude that $\partial15N$ stable isotope values reflect microbial community composition, which differ between drained and natural peatlands.

## 1 Introduction

Even though peatlands cover only 3-4 % of the Earth's land surface (Leifeld and Menichetti, 2018), they act as an enormous

sink for greenhouse gases in natural conditions (Yu et al., 2011; Joosten, 2008). In Europe 70% of the peatlands are currently degraded (Joosten and Couwenberg, 2001). These degrading peatlands account for five percent of the anthropogenic $CO_2$ emission (Leifeld and Menichetti, 2018; Zedler and Kercher 2005). Despite this dramatic peat decline, we lack reliable and transferable tools providing time- and cost-efficient information of peatland condition. We hypothesize that nitrogen (N) stable isotopes could serve as such a tool.

In natural peatlands most biological metabolism and nutrient cycling takes place in the acrotelm (uppermost aerobic peat horizon with living vegetation) (Asada et al., 2005; Artz, 2013). In the water-saturated catotelm (deeper, anaerobic horizon) organic substrates are decomposed at much smaller rates owing to anoxic conditions and low pH values (Asada et al., 2005; Artz, 2013; Lin et al., 2014). In the mesotelm, a peat horizon situated between acrotelm and catotelm, water table levels and oxygen content constantly fluctuate, resulting in shifting oxic and anoxic conditions and shifting metabolism processes

(Asada et al., 2005; Artz, 2013; Lin et al., 2014). Clymo and Bryant (2008) therefore defined the mesotelm as a "transition horizon".

Stable C and N isotopes are correlated with vegetation composition and microbial decomposition processes. As decomposition induces an enrichment of heavy isotopes (15N, 13C), vegetation is mostly more depleted of 15N and 13C than microbial and recycled substrate. Alewell et al. (2011) and Krüger et al. (2014) reported distinct changes in $\partial13C$ values

for palsa peat with the onset of decomposition of hummocks and various authors observed the same trend with decomposition in peatlands of other climate conditions (Krüger et al., 2016a; Novak et al, 1999; Hobbie et al., 2017; Biester et al., 2014). The distinct $\partial13C$ depth pattern is a consequence of the use of different sources by fungi and bacteria as investigated by Kohl et al. (2015) for peat profiles. They conclude that an increasing $\partial13C$ signal is caused by differences in biomass synthesis and carbon sources used by fungi and bacteria, which was also reported by Lichtfouse et al. (1995).

We suggest that changing microbial abundance must also be reflected in specific nitrogen stable isotope depth patterns. But, in contrast to the well-studied carbon isotope depth pattern, there are less data available for nitrogen. Caused by changing microbial abundance, which were also mentioned by Tfaily et al. (2014) or Hobbie and Ouimette (2009), we hypothesize that a peak of $\partial15N$ values will be present at the same depth where the maximum diversity of microbial metabolism can be found.

To indicate the abundance of specific microbial communities in the peat horizons phospholipid fatty acids (PLFAs) are valid markers, because they are specific and persistent compounds of cell membranes of different species (Willers, Jansen van



Rensburg and Claassens, 2015; Reiffarth et al., 2016). Therefore PLFAs enable us to make qualitative and quantitative statements about the relative abundance of different microbial communities.

Here, we use specific terms for the change points in the stable isotope depth pattern and the horizon descriptions. The points

where the stable isotope signals undergo a sudden directional shift with depth are called "turning points" according to Alewell et al. (2011). Furthermore, the mesotelm is enlarged in drainage-affected cores and we will use the term "upper mesotelm" for the uppermost part of a drainage-affected horizon and the term "lower mesotelm" for the deeper part of a drainage-affected horizon. The bottom of the drainage-affected horizon and the onset of the underlying catotelm are marked by the ∂13C turning point. If rewetting processes are present above the mesotelm, the horizon is called "rewetting horizon".

Our aim is to evaluate ∂15N depth trends as indicators of specific peatland conditions and to study whether ∂15N depth trends of natural and drainage-affected sites indicate, in parallel to ∂13C, a shift in dominant microbial communities, reflected by specific PLFAs. We are also interested to see if there are distinct changes of ∂13C and ∂15N isotope signatures with the onset of rewetting processes. We match isotope depth trends (∂13C, ∂15N) with bulk density (BD) and carbon/nitrogen ratio (C/N). BD acts as an indicator for decomposition, because decomposition processes are leading to

higher compaction of the peat soil and therefore increasing BD values. The C/N ratio gives us also information about the degree of decomposition. With increasing decomposition the C/N ratio decreases. With an additional microscope analysis of the macro-residuals in the peat horizon, we will get information of the humification indices (HI) and the vegetation assemblages. To test our hypothesis of changing dominant microbial communities as drivers for isotope patterns, we do a PLFA analysis of four investigated sites – two drainage-affected and two natural sites in Degerö Stromyr and Lakkasuo. We

will test the existence of two Gram positive - bacterial (i-C15:0; a-C15:0) markers and one fungal (C18:2ω9c) marker (Sundh, Nilsson and Borga, 1997; Elvert et al., 2003). In the Swedish site Degerö Stormyr we add information of tree ring development as an indicator of peatland dynamics.

## 2 Material and methods

### 2.1 Site description

We studied five oligotrophic peatlands (Tab. 1, Tab. 2).

Degerö Stormyr (200 m above see level (a.s.l.)) is situated in Northern Sweden, at the Kulbäcksliden Experimental Forest near Vindeln, between the rivers Umeälven and Vindelälven (Eurola, Hicks & Kaakinen, 1984). It is an acidic mire with minerotrophic conditions and consists of interconnected small mire patches divided by ridges of glacial till. The climate is characterized as cold with no dry seasons and cold summers (Dfc-zone after Köppen-Geiger classification; Peel et al., 2007).

In Degerö ditches were installed at the beginning of the 20th century, were closed in 2017 and a naturally reestablishment of sphagnum took place afterwards. The water table is at the surface in the natural part (DNM) (Nielsson et al., 2008) and in around 10-15 cm depths at the drainage-affected location (DDC).



Lakkasuo (150 m a.s.l.), Central Finland, is an eccentric peatland complex with two parts. In the southern part the conditions are ombrotrophic, whereas the northern part is minerotrophic (Minkkinen et al. 1999). Lakkasuo is also located in the cold

climate zone, with no dry seasons and cold summers (Dfc-zone after Köppen-Geiger classification; Peel et al. 2007). The 1961 installed ditches (70 cm depth, spacing of 40 m – 60 m) affect approximately 50 % of the peatland (Minkkinen et al., 1999). In the ombrotrophic natural site (LON) the water table was around 13 cm below ground surface. The ombrotrophic drained site (LOD) had a water table of 26 cm depth (average), whereas the water table is near the surface at the minerotrophic natural site (LMN) and in an average depth of 36 cm in the minerotrophic, drained site (LMD) (Minkkinen et

al., 1999) (Tab. 1, Tab. 2).

In the Black Forest three mires were investigated: Breitlohmisse, Ursee and Rotmeer. They are located in the temperate climate zone with no dry seasons and warm summers (Cfb-zone after Köppen-Geiger; Peel et al., 2007). In the mires of the Black Forest ditches were installed in the middle of the 20th century. Breitlohmisse (810 m a.s.l., 50 km southeast of Baden-Baden) is minerotrophic and is located in the Northern part of the Black Forest. The mire is mostly lanced with ditches for

huntsman ships (Br2). The ditches are naturally refilled with Sphagnum. The water table is at an average depth of 15 cm in the natural center (Br1, Br2), and is found at lower depths near the degraded edges of the mire (Br3, Br4). Rotmeer (960 m a.s.l., 40 km southeast of Freiburg i.B.) and Ursee (850 m a.s.l., 45 km southeast of Freiburg i.B.) are both in the Southern Black Forest. Rotmeer consists of an ombrotrophic center (Ro1) (water table at the surface), surrounded by a minerotrophic part with signs of decomposition (Ro2, water table around 12 cm depth) and without mosses at the edges (Ro3, water table

below 12 cm depth). Urmeer is minerotrophic. A quaking bog forms the center with the water table at the surface (Ur2), whereas the edges had a lower water table (Ur1) (Tab. 1, Tab. 2).

**2.2 Soil sampling and bulk analyses**

In May 2012 (Breitlohmisse), June 2012 (Rotmeer), July 2012 (Ursee) and September 2013 (Degerö and Lakkasuo) three volumetric peat cores were drilled per site with a Russian peat corer (Eijkelkamp, The Netherland) at a medium stage of

small-scale topography. In Degerö cores were sampled in the assumed natural center of the mire (DNM) and in one-meter distance to a drainage ditch (one meter depth) (DDC). In Lakkasuo we took cores at the natural sites (ombrotrophic natural (LON), minerotrophic natural (LMN)) and the drainage-affected locations (ombrotrophic drained (LOD), minerotrophic drained (LMD)). For Ursee two cores were taken, one in the natural center (Ur2) and one at the drainage-affected edge of the mire (Ur1). In Breitlohmisse and Rotmeer we took cores in a transect from natural (Br1, Ro1) to strong drainage-affected

(Br4, Ro3) sites. Each core has a composite length of one meter. Here, we focus on the uppermost 60 cm because this part included the drainage-affected horizon and no major changes in isotopic composition were observed at the natural sites below the acrotelm.

Directly after drilling HI were determined for each horizon with the von Post scale. The von Post scale has a range form 1 to 10. HI 1 indicates natural condition with undecomposed, completely visible vegetation residuals. HI 10 represents a strongly

decomposed horizon without visible vegetation residuals. (Silc and Stanek, 1977)



The cores were encased in plastic shells and covered with plastic wrap, stored in coolers, and transported to the laboratory. The cores were sliced in 2 cm sections and every second layer was analysed, giving a 4 cm depth resolution. Samples were oven-dried at 40 °C for 72 h, and homogenized with a vibrating ball mill (MM400, Retsch, Germany). Stable C and N isotopic signatures were measured an elemental analyser combined with an isotope ratio mass spectrometer (EA-IRMS)

(Inegra2, Sercon, Crewe, UK). Carbon isotopic composition (13C/12C) was expressed relative to Vienna Pee-Dee Belemnite (VPDB) standard and reported in delta notation (‰), stable nitrogen isotopes were expressed relative to the atmospheric nitrogen standard and reported in delta notation (‰). C/N was determined with the mass relationship of the measured bulk content of C and N. Bulk density was measured with volumetric samples, which were weighted before and after drying.

In Degerö tree rings of seven individual trees were analysed (Pinus sylvestris) to obtain information of growth conditions

and to enhance therefore our knowledge of drainage history.

### 2.3 Fatty acid analysis

Four cores (per site one drainage-affected and one natural core) were selected to do a fatty acid analysis: two sites in Lakkasuo, LOD 1 and LON 3 and two sites for Degerö Stromyr, DDC 3 and DNM 1. We took subsamples in all cores in the acrotelm (respectively at the end of the mesotelm in DDC) and in the catotelm. At the drainage-affected sites DDC 3 and

LOD 1 we took also samples in the middle and at the end of the mesotelm.

We processed 0.2 – 1.1 g of sample for the phospholipid extraction with a mixture of CH2Cl2 : MeOH (9:1 v/v) in an Accelerated Solvent Extractor (Dionex ASE 350). 50 µl of an internal standard with nonadecanoic acid was added before processing each sample.

The total lipid extracts (TLE) were saponified by adding 2 ml of KOH dissolved in MeOH (12%) and putting it in the oven

for 3 hours at 80°C.

Following the method of Elvert et al. (2003) TLE was afterwards pooled with 1 ml KCl (0.1 mol) and the neutral fraction was extracted by agitating three times with hexane. Neutral fraction in the supernatant was separated, dried under a stream of N2, and stored in the fridge for later analysis. We acidified the rest of the TLE with fuming hydrochloric acid to a pH of 1. The acid fraction was extracted by agitating again three times with hexane. The acid fraction in the supernatant was

separated and hexane was reduced to almost dryness under a stream of N2. Then the acid fraction was methylated by adding 1 ml Boron-Trifluoride (BF3) in MeOH (12-14%) and putting it in the oven for 1 hour at 60°C. Afterwards the PLFA fraction was pooled with KCl (0.1 mol) and transferred in 2 ml vials by agitating three times with hexane. The PLFAs were quantified with a Trace Ultra gas chromatograph (GC) equipped with a flame ionization detector (FID) (Thermo Scientific, Waltham, MA, USA). The carrier gas (helium) had a constant flow of 1.2 ml per minute and the GC-FID was set to splitless

mode. Detector temperature was 320°C and the samples (dissolved in hexane) were injected by 300°C. The starting temperature of the oven was 50°C. The temperature was increased by 10°C per minute to 140°C. The temperature was held for 1 minute before it was increased up to 300°C. This temperature was held for 63 minutes.





To identify the fungal and bacterial markers, we used the Bacterial Acid Methyl Esters standard (BAME, Supelco Mix), which includes as markers the PLFAs i-C15:0 and a-C-15:0 for Gram positive - bacteria (Zelles, 1997, O`Leary and

Wilkinson, 1988; Vestal and White, 1989) and C18:2ω9c for fungi (Andersen et al., 2010; Sundh, Nilsson and Borga, 1997; Zelles, 1997, O`Leary and Wilkinson, 1988; Vestal and White, 1989). Quantification of the PLFAs was done using the internal standard, C19:0 FA, after correcting for the methyl group, added during methylation reaction.

**2.4 Data evaluation and statistical analysis**

As we were interested in comparing the depth trends of all single profiles with each other, we first normalized the depths of

the cores. This was done using the depth of the ∂15N turning point (see chapter 3.1) in each drainage-affected profile as the anchor point serving as normalized depth (normD). The normalized depth of this anchor point was set to 20 cm depth (normD = 20 cm, fig. 1) in each single core. In the corresponding natural cores, we have transferred the values from the same depth related to the drained core into the same norm depth. For example the values of the natural site (DNM) in depth of 13 cm (depth of the turning point of ∂15N in the corresponding DDC core) were set to 20 cm normD.

In a second step, because we were mainly interested in trends and not the absolute values, we normalized the isotopic values themselves, because the range of ∂15N varied considerably between the sites, whereas the trends show consistent patterns (fig. 1). Therefore, to be able to do a meaningful comparison we set therefore the value of ∂15N at the turning point to zero in each profile:


$$normalized\ \partial^{15}N\ [‰] = \partial^{15}N\ [‰] - \partial^{15}N\ [‰]\ at\ turning\ point$$

Using the same procedure, all other parameters (∂13C, C/N, BD) were normalized using the same anchor point (e.g., ∂15N turning point):


$$normalized\ value\ (\partial^{13}C\ [‰],\ BD,\ C/N) = value\ (\partial^{13}C\ [‰],\ BD,\ C/N) - value\ (\partial^{13}C\ [‰],\ C/N,\ BD)\ at\ \partial^{15}N\ turning\ point$$

Using the above procedures means to decide on the depth of the ∂15N turning points, which we backed up statistically with a t-test (p ≤ 0.05) and an integrated change point analysis with the software package "changepoint" in R (version 1.0.153). These analyses were done for each of the drained sites separately and also in addition with an average of all locations. For

the t-test, we analyzed for each depth if ∂15N values in the drainage-affected horizon are of the same population as the values of the natural sites (H0: drained and natural values are of the same population). For the changepoint analysis, the variance of ∂15N was evaluated with a linear gradient over the whole drainage-affected peat profile against the variance of three/ four separated linear gradients (rewetted part (if present), upper mesotelm, lower mesotelm, catotelm). Here, we define

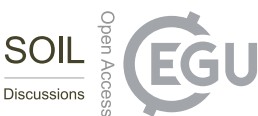

the starting point of the drainage-affected horizon with the onset of a shift in the $\partial 15N$ values upward and the end of this

horizon with the stabilization of the $\partial 15N$ values towards the surface.

We also determined the slopes of each single core to get information on the strength of differences of the isotopic values with depth. First, the whole peat profile of each drainage-affected core was analysed as one trend (called "overall profile"). Second, profiles were separated into different horizons: (i) rewetted horizon (if present), (ii) upper mesotelm, (iii) lower mesotelm and (iv) catotelm. If values were clearly changing with depth slopes were closer to zero. In horizons with

stabilized values slopes were distinct higher or lower zero.

In the following we present only the normalized data. Raw data without normalization are available in the supplementary information.

### 2.5 Tree Ring width and microscope analysis of peat

The investigation of the tree ring width of seven surrounding trees (Pinus sylvestris) in Degerö Stormyr was done with a

hand-operated wood driller (Djos/ Sweden, 5 mm diameter). Samples were fixed on wooden carriers. The tracheids (elongated cells of the xylem of vascular plants) were cut with a sharp carbon blade and analysed with an impinging light binocular (60x – 160x amplification).

Peat samples of four study sites were analysed using an impinging light binocular (60x – 160x amplification) to get an overview of the vegetation assemblages and to differentiate horizons. For detailed information (distinction of Sphagnum

species) the samples were elutriated with water, pigmented with methyl-blue and analysed under a transmitted light microscope (100x – 640x amplification).

## 3 Results and discussion

### 3.1 Stable nitrogen isotope depth trends as indicators for peatland conditions

While mineral soils have been shown to have continuous increasing values of $\partial 15N$ (Nadelhoffer et al, 1996, Högberg et al,

1997), we found increasing $\partial 15N$ and $\partial 13C$ values with depth down to particular, isotopic specific turning points in drainage-affected peatland soils (fig. 1). We defined three types of peatland conditions from the observed depth trends of $\partial 15N$: (a) natural, (b) drainage-affected up to the surface, (c) profiles with a rewetted horizon above the drainage-affected horizon. (fig. 1)

Eight out of nine studied drained peatlands and the average trend confirm the existence of a $\partial 15N$ turning point, with a p-

value $< 0.05$ for the difference in $\partial 15N$, in the center of the mesotelm in contrast to the $\partial 15N$ values in undisturbed horizons (Tab.3), with a non-significant difference for one drained site: Breitlohmisse (Br3). The latter was most likely related to generally higher $\partial 15N$ values of the natural site in Breitlohmisse (Br1) compared to a smaller increase of $\partial 15N$ in this





drainage-affected site (Br3). The depth of ∂15N turning point (center of the mesotelm) differs from ∂13C turning point (end of the mesotelm) for all investigated sites (fig. 2).

Changed slope values of the separated horizons indicate significant trend changes (Tab. 5). In anaerobic conditions (natural, catotelm) with stabilized isotopic values with depth, slopes were distinct different to 0 [cm/‰]. ∂15N values seems to change within the mesotelm rapidly and slope values were closer to zero. Most interesting was a switch to negative trend values at the ∂15N turning point in all investigated drainage-affected sites, which marks the beginning of the lower mesotelm. (Tab. 5)

In natural conditions (type (a)), all investigated parameters had a low variability and indicated natural, wet mire conditions (fig. 1). There were two exceptions: Breitlohmisse 1 (Br1) (40 - 60 cm normD) and Rotmeer 1 (Ro1) (30 – 50 cm nromD), with trend instabilities of ∂15N. This might indicate some minor drainage or disturbance in the wetland sites we classified as "natural" (fig. 1).

In contrast, the values of the drained sites show significant trends. We found two different trend types in the drainage-
affected sites: Type (b) and (c) (fig. 1). For type (b) we distinguished six sites: Lakkasuo ombrotrophic drained (LOD), Breitlohmisse 2 (Br2), Breitlohmisse 3 (Br3), Breitlohmisse 4 (Br4), Rotmeer 2 (Ro2) and Rotmeer 3 (Ro3) with clear signs of decomposition up to the surface. Type (c) was visible in three sites: drainage-affected site Degerö Stromyr (DDC), minerotrophic drainage-affected site Lakkasuo (LMD) and Ursee 1 (Ur1). At type (c) sites the isotopic values, C/N and BD were stabilized again above the mesotelm. Therefore, they are assumed to be in a "new" natural status (fig. 1, fig. 2).

Below 8 cm (normD, average profile) all drainage-affected profiles showed the typical signs of the upper mesotelm with increasing values of ∂15N, ∂13C and BD, down to the ∂15N turning point, and decreasing C/N. Below the ∂15N turning point, in the lower mesotelm, ∂15N values were decreasing. In this horizon ∂13C values, C/N and BD were increasing. The end of the lower mesotelm was mostly linked to a clear shift in ∂13C trend to either stable values or a slow decreasing trend; hence, we called this point ∂13C turning point (28 cm normD, average profile) (e.g. Krüger et al. 2014). Constant C/N, BD
and ∂15N values below the ∂13C turning point served also as indicators for reduced compaction and decomposition. Most likely the ∂13C turning point marked the onset of permanent waterlogged anaerobic conditions (e.g. Krüger et al. 2016a). The similarity in trends in these deeper parts of the drainage-affected sites to those of the catotelm in the natural sites supported the assumption of an intact catotelm below the ∂13C turning point (fig. 1, fig. 2.). (For the single ∂13C, C/N and BD values of all peat cores see supplementary information).

**3.2 Depth profile of vegetation assemblage and water table in connection with isotope depth pattern**

All sites, which we attributed as "natural" (type (a)), had a water table near the surface (<10 cm), and macro-residuals were highly visible throughout the profile, HIs were low and the main living vegetation was Sphagnum spp. (tab. 6, tab.7).

All drainage-affected sites had higher HIs even if no direct modifications in the vegetation assemblage could be documented. For type (b), there was little or no Sphagnum visible at the surface and the water table was found at lower depths. Macro-
residuals were more affected by decomposition and HIs were high up to the surface. These results were in line with our





interpretation of isotope signatures of a drainage-affected horizon up to the surface. Especially the ombrotrophic-drained site (LOD) was influenced by drainage. Here Sphagnum species seem to have disappeared and were replaced by mosses of drier environments or mosses were completely absent (tab. 7).

For type (c), vegetation assemblages were mainly composed of Sphagnum spp. and the water table was near the surface. HIs

were low in the rewetted horizon and macro-residuals were preserved well (tab. 6, tab. 7). With the onset of the upper mesotelm, HIs and decomposition of macro-residuals was high. In the lower mesotelm, the HIs were decreasing and more macro-residuals were visible. In the catotelm, the quality of macro residuals was higher than in the mesotelm and the HIs were even lower.

### 3.3 Tree ring width are verifying isotope signals of changing peatland conditions

Tree ring width is a marker for the wellbeing and/or growth rate of trees. Young trees have a small scope coupled with high growth rates, which leads to thicker tree rings. Tree rings get smaller with increasing age of the tree. If there are no environmental stressors like heat, increasing wetness or drought, tree rings are bigger and the cell lumen is higher compared to trees at sites with environmental stress. With increasing environmental stress tree ring width decreases (Stoffel et al., 2010). Before 1992, tree rings at the drainage-affected site (DDC) site showed only a slightly decreasing trend, which could

be due to aging (average of 1.3 mm width in the 1930s to an average width of 0.9 mm in the late 1980s). The draining ditches in Degerö Stormyr were established in the beginning of the 20th century, which supports these results, with dryer and therefore better growth conditions for trees. From 1992 onwards tree ring widths decreased, reaching 0.2 mm in 1998 and thereafter. These results were concurrent with the isotope analysis, because both suggest a restoration of natural wet peatland conditions. Rewetted conditions are no longer suitable for trees and lead to smaller tree ring width according to adverse

environmental conditions for tree growth. These findings underpin our suggestion of rewetted conditions at this site in Degerö.

### 3.4 Linkage of microbial abundance and isotopic signature

At the natural sites, all samples near the surface had relatively low, fungal-derived PLFA concentrations. In the catotelm the values were very low, dominated by bacterial-derived PLFAs (fig. 3).

In the drainage-affected sites microbial-derived PLFA abundance was increased over the whole mesotelm. This pattern could be caused by the improved conditions for metabolism processes by drainage: enhanced oxygen abundance and relatively high nutrient availability of the prior conserved plant material. In the upper mesotelm dominantly fungal-derived, but also bacterial-derived, PLFAs were visible. At the $\partial15N$ turning point lowered values of fungal markers and increased bacterial-derived PLFAs could be found. In the lower mesotelm the abundance of microbial-derived PLFAs was generally decreased,

however more bacterial than fungal marker were detected. In the catotelm microbial-derived PLFA abundance was low, similar to the natural sites. (fig. 3)





The observed switch, from predominantly fungal abundance in the acrotelm to bacterial abundance in the lower mesotelm, is in agreement with Kohl et al., 2015 and Schmidt and Bölter, 2002. Also Andersen et al (2013) stated out, that fungi biomass is decreasing in peatland soils. Our investigations indicate, that this switch is linked to the depth pattern of isotopic N (fig.

4), which is in accordance with the findings of Wallander et al. (2009); Winsborough and Basiliko (2010) and Myers et al. (2012). With increasing depth microbial utilization of nitrogen and carbon via alternative pathways and from recycled sources (with enriched $\partial 15N$ values) is necessary (Dijkstra et al. 2006, Dröllinger et al. 2019). The reason for this is that increasing depth leads to increasing oxygen limitation and thus declining decomposability of the remaining substrate.

The nitrogen isotopic signal of peat material is generally depleted compared to atmospheric nitrogen (which is, per

definition, 0 ‰, if air is used as the nitrogen isotopic standard). As such, the average signal of the relatively undecomposed peat (e.g., the "natural" sites, the catotelm or the peat grown under rewetted conditions) is -10 to -4 ‰. The latter is parallel to the isotopic depletion of $\partial 13C$ in plant material, due to the general preference of plants for the lighter isotopes 12C and 14N. However, with the onset of drainage, decomposition of the organic plant material also below the acrotelm take place, resulting in a preferential mineralization of 14N and an enrichment of 15N in the remaining peat material. In acid bogs under

aerobic conditions, fungi will dominate the general metabolism (Thormann et al., 2003). This is pictured by the highest amount of fungal-derived PLFAs in the acrotelm and the upper mesotelm. Fungi are preferred decomposers of primary plant material (Wallander et al., 2009; Thormann et al., 2003) hence the depleted plant isotopic signal is relatively preserved in the upper most aerobic horizons. Furthermore, fungi have a relatively low nitrogen demand compared to bacteria (Myers et al. 2012). This explains, in line with Thormann (2005), our pattern of lower $\partial 15N$ isotope values in the acrotelm compared to

the mesotelm, where the incoming plant/ moss signal is more and more enriched with depth by decomposition processes and nitrogen turnover. With increasing depth and increasing oxygen limitation fungal metabolism decreases (Thormann, 2011). In parallel, bacterial metabolism increases as Lin et al. (2014); Hu et al. (2011) and Bauersachs et al. (2009) reported. They found evidence for bacterial-dominated decomposition in hypoxic conditions. Bacterial metabolism is generally faster than fungal metabolism (Brunner et al., 2013). Faster turnover is connected with less isotopic fractionation. In addition, bacterial

metabolism needs higher amounts of nitrogen. In summary, with increasing depth and increasing bacterial metabolism most of the available nitrogen will be immobilized, which results in no or low fractionation of the bulk material (e.g., no preferential loss of the lighter 14N). Hence, the $\partial 15N$ turning point could be caused by the N-limitation of peatland ecosystems with low oxygen availability. At this depth (20 cm normD) bacteria and fungi compete most over decomposable substrates (not necessarily nitrogen), resulting in the highest turnover rates with enrichment of $\partial 15N$ in the remaining peat,

as we know from mineral soils with aerobic decomposition (Alewell, et al. 2011, Nadelhofer, et al 1996). Tfaily et al. (2014) also reported the highest $\partial 15N$ values within the mesotelm. This pattern is reflected in the equilibrium of fungal- and bacterial-derived PLFAs at the $\partial 15N$ turning point (fig. 3).

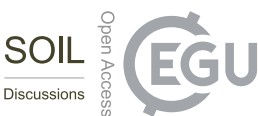

In the lower mesotelm, oxygen limitation increases, leading to decreasing microbial metabolism and decreasing concentrations of microbial-derived PLFAs. The decreasing microbial metabolism leads to simultaneously decreasing $\partial 15N$

values because an increasing amount of undecomposed vegetation (with low $\partial 15N$ values) will be conserved. (fig. 4)

Finally, with the establishment of permanently waterlogged anaerobic conditions (indicated by the $\partial 13C$ turning point), PLFA concentration decreases sharply. In the catotelm decomposition processes are mostly inhibited, which leads to stable $\partial 15N$ and $\partial 13C$ values, close to the original vegetation signals (Alewell et al., 2011; Krüger et al, 2015). (fig. 1, fig3)

### 4. Conclusion

Our results show significant differences in the nitrogen isotopic depth trends of natural, drainage-affected and rewetted peat profiles. We validated our isotopic hypothesis with microscope analysis of the vegetation remains in the cores as well as the investigation of tree rings as indicators for changing hydrological conditions in the past. An analysis of bacterial- versus fungal- derived PLFAs unravelled the changing microbial abundance with depth as characterized by high fungal abundance in the aerobic acrotelm with low nitrogen demand and turnover; transition to a mixture of decreasing fungal and increasing

bacterial abundance in the upper mesotelm, competing on organic substrates, resulting in an enrichment of $\partial 15N$; decreasing microbial decomposition, dominated by bacterial abundance, in the lower mesotelm and impeded microbial metabolism and stabilized $\partial 15N$ values in the anaerobic catotelm.

Carbon isotope signatures are also changing with drainage, but there is neither a clear indication of a switch in microbial abundance within the drainage-affected horizon, nor a clear change in the trend with rewetting of the peatland, as it is visible

for the nitrogen signatures due nitrogen limitation and recycling processes. Summing up, $\partial 15N$ depth profiles in peat might give more insights into the degree of microbial transformation, because they reflect more precisely different microbial abundance than carbon isotope signatures do. Therefore, we conclude that $\partial 15N$ depth profiles could act as a reliable and efficient tool to get fast and easy information about peatland status, restoration success and drainage history.

**Author contribution**

Christine Alewell and Jens Leifeld are the supervisors of the project. Miriam Groß-Schmölders, Jan Paul Krüger, Axel Birkholz and Kristy Woodard were doing the measurements. Pascal von Sengbusch was doing the microscopy and vegetation analysis. Miriam Groß-Schmölders prepared the manuscript with contribution of all co-authors.





**Competing interests**

The authors declare that they have no conflict of interest.

**Acknowledgements**

This study was founded by the Swiss National Science Foundation (Project No. 169556).

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



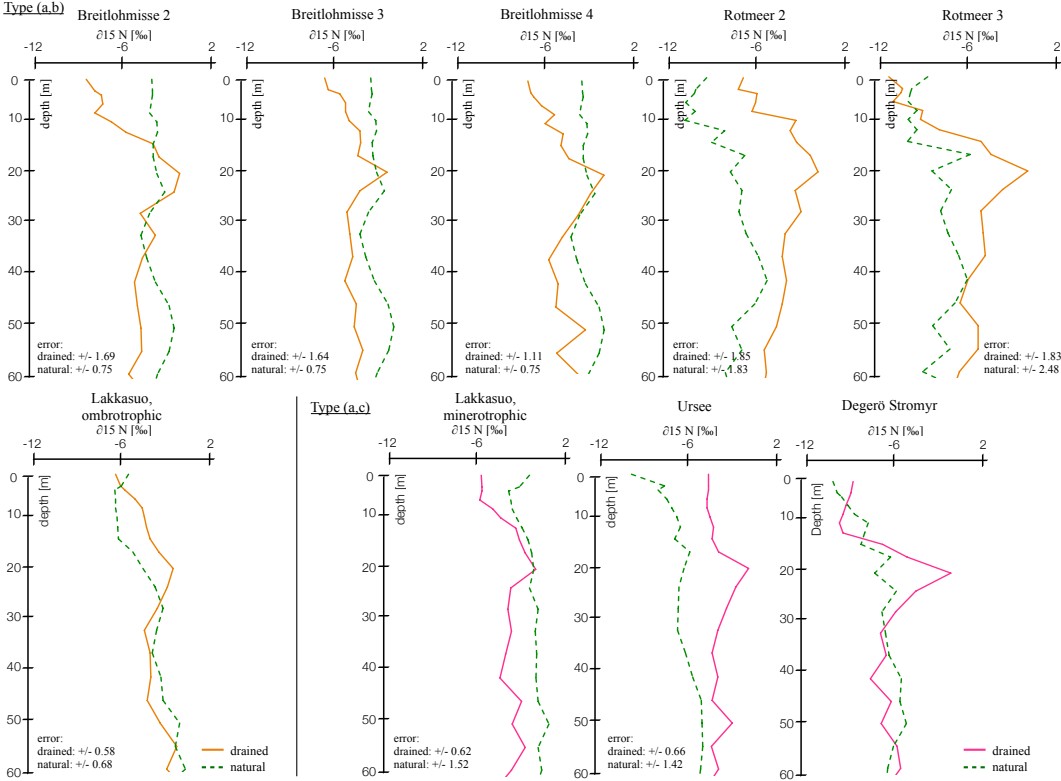

**Figure 1:** $\partial^{15}$N depth profiles in all natural and drained (or rewetted) sites; with normalized depth and normalized $\partial^{15}$N values (see chapter 2.4); trend types: (a) natural (green), (b) drainage-affected up to the surface (orange) and (c) rewetted above drainage (pink) (For single, non-normalized values see supplementary information).





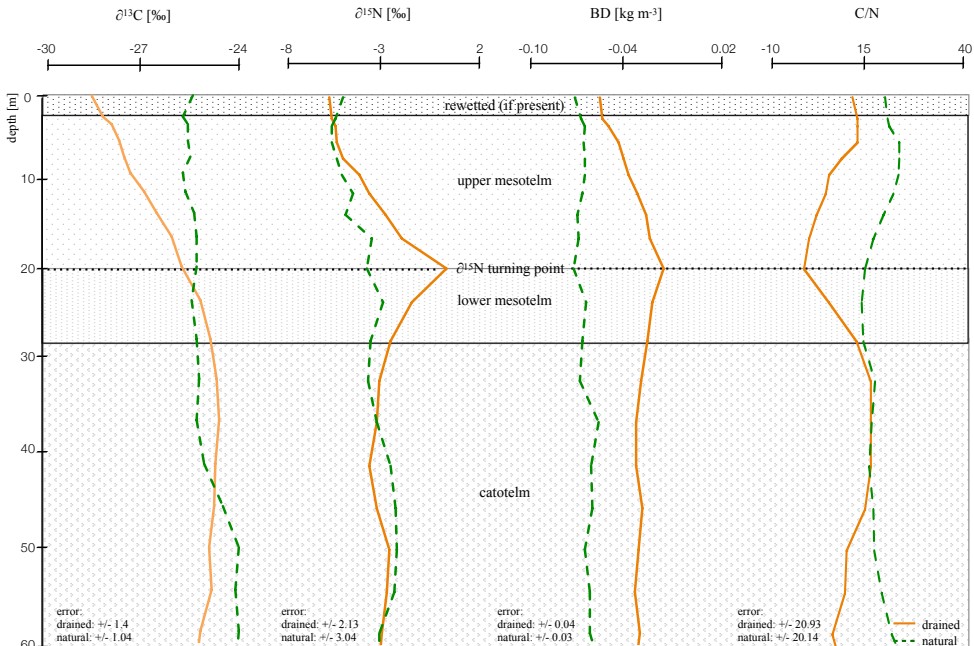

**Figure 2: Mean depth trends ($\partial^{15}$N, $\partial^{13}$C, C/N and BD) of natural and drained sites of all nine investigated peatlands with normalized depth and normalization based on $\partial^{15}$N signatures (see chapter 2.4; For single $\partial^{13}$C, C/N and BD values of all peat**
**cores see supplementary information).**



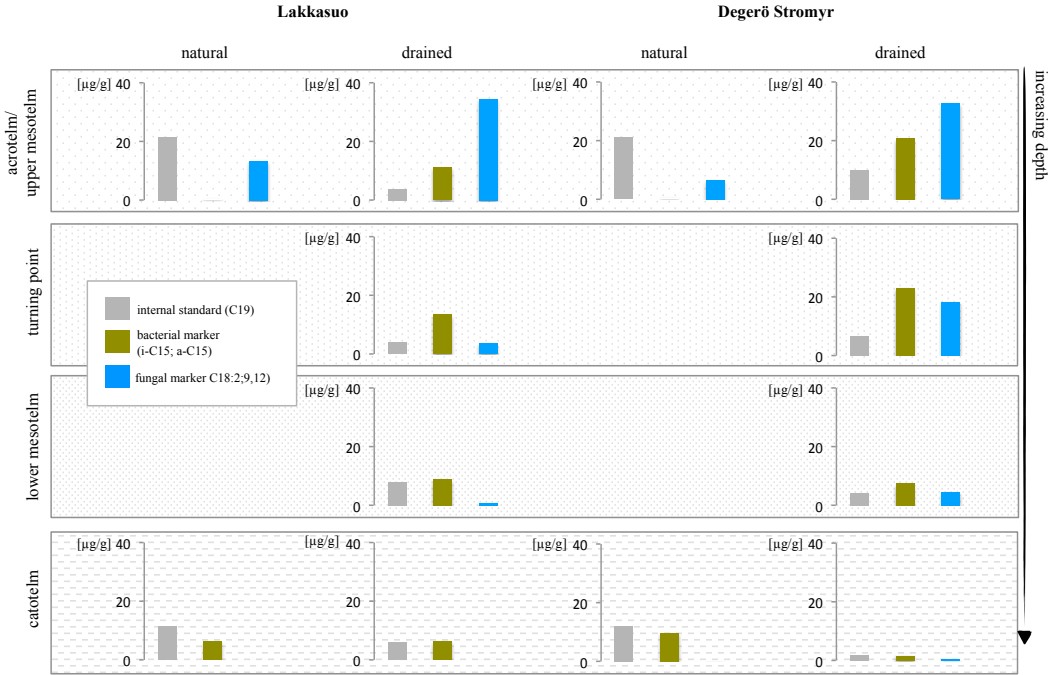

Figure 3: PLFA amount of bacterial and fungal marker in LOD, LON, DDC and DNM in different horizons.





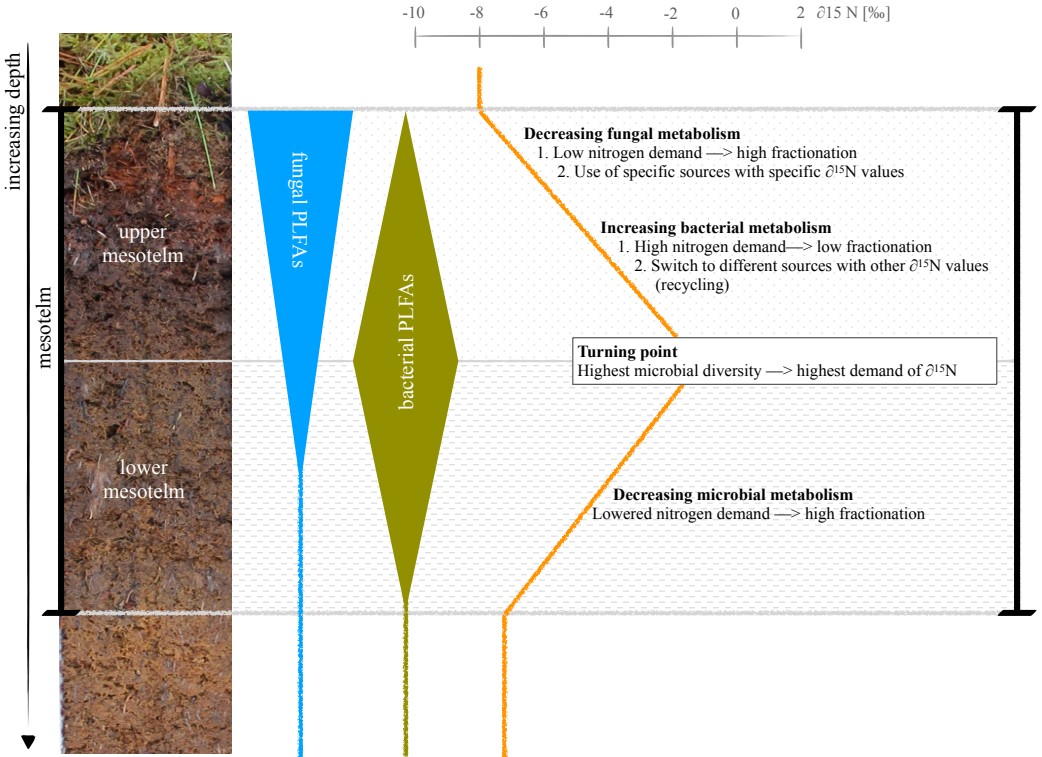


**Figure 4: Hypothesis of a microbial switch (fungi to bacteria) with depth, reflected by specific PLFAs, and its influence of the $\partial^{15}$N depth trend; example photo and $\partial^{15}$N values of the ombrotrophic, drained site in Lakkasuo (LOD) (note all isotope values are normalized to zero at turning point).**




**Table 1: Labeling of all drilling sites**

| Location | Labeling |
|---|---|
| Degerö | |
| natural mire | DNM |
| drained | DDC |
| Lakkasuo | |
| minerotrophic natural | LMN |
| minerotrophic drained | LMD |
| ombrotrophic natural | LON |
| ombrotrophic drained | LOD |
| Breitlohmisse | |
| natural mire | Br1 |
| natural dry | Br2 |
| drained | Br3 |
| near the mire edge | Br4 |
| Rotmeer | |
| natural mire | Ro1 |
| drained, with Sphagnum | Ro2 |
| drained, without Sphagnum | Ro3 |
| Ursee | |
| natural mire | Ur2 |
| drained | Ur1 |



**Table 2: Overview of studied mires; coordinates (lat./long.); mean annual temperature (MAT); annual precipitation (P);**
**Sphagnum mosses (Sph.) (Laine et al., 2004; Nielsson et al., 2008; DWD 2018, Alexandersson et al., 1991; Armbruster et al., 2003)**

| Country | Mire | lat/long.. | MAT | P | Main vegetation on top | |
|---|---|---|---|---|---|---|
| | | | [°C] | [mm] | natural | drained |
| Sweden | *Degerö Stromyr* | 64°11'lat., 19°33'long. | +1.2 | 523 | Sph. majus | Sph. balticum |
| Finland | *Lakkasuo* | 61°48'lat., 24°19'long. | +3 | 700 | Sph. angustifolia | Sph. angustifolia |
| Germany (Black | *Breitlohmisse* | 48°41'lat., 8°25'long. | +7 | 835 | Sph. capilifolium | Sph. capilifolium |
| Forest) | *Ursee* | 47°51'lat., 8°25'long. | +7 | 1600 | - | - |
| | *Rotmeer* | 47°52'lat., 8°6'long. | +7 | 1600 | Sph. rubellum | Sph. rubellum patches |





**Table 3: Level of error probability according to T-tests of $\partial^{15}N$ measurements to identify significant differences (p<0.05, marked in red) between the drainage-affected and the natural sites; black border: normalized turning point of $\partial^{15}N$, drained sites: Degerö (DDC), Lakkasuo (ombrotrophic (LOD), minerotrophic (LMD), Rotmeer (Ro2, Ro3), Ursee (Ur1), Breitlohmisse (Br2, Br3, Br4)**

| depth | all | DDC | LOD | LMD | Ro2 | Ro3 | Ur1 | Br2 | Br3 | Br4 |
|---|---|---|---|---|---|---|---|---|---|---|
| 2 | 0.01 | 0.18 | 0.38 | 0.01 | 0.21 | 0.28 | 0.00 | 0.12 | 0.03 | 0.06 |
| 4 | 0.00 | 0.25 | 0.85 | 0.07 | 0.19 | 0.19 | 0.36 | 0.17 | 0.30 | 0.04 |
| 5 | 0.03 | 0.52 | 0.13 | 0.25 | 0.34 | 0.23 | 0.66 | 0.20 | 0.59 | 0.06 |
| 7 | 0.28 | 0.89 | 0.03 | 0.25 | 0.08 | 0.34 | 0.77 | 0.09 | 0.72 | 0.04 |
| 8 | 0.50 | 0.40 | 0.00 | 0.35 | 0.12 | 0.88 | 0.83 | 0.04 | 0.83 | 0.15 |
| 10 | 0.79 | 0.06 | 0.00 | 0.40 | 0.05 | 0.99 | 0.97 | 0.13 | 0.55 | 0.07 |
| 12 | 0.55 | 0.28 | 0.00 | 0.70 | 0.34 | 0.94 | 0.34 | 0.11 | 0.77 | 0.06 |
| 14 | 0.02 | 0.16 | 0.00 | 0.47 | 0.26 | 0.28 | 0.02 | 0.99 | 0.63 | 0.02 |
| 17 | 0.02 | 0.35 | 0.00 | 0.60 | 0.01 | 0.48 | 0.10 | 0.33 | 0.13 | 0.61 |
| 20 | 0.00 | 0.02 | 0.01 | 0.21 | 0.07 | 0.05 | 0.03 | 0.03 | 0.12 | 0.06 |
| 24 | 0.05 | 0.26 | 0.05 | 0.05 | 0.39 | 0.25 | 0.05 | 0.34 | 0.24 | 0.59 |
| 28 | 0.61 | 0.42 | 0.98 | 0.01 | 0.12 | 0.41 | 0.08 | 0.40 | 0.35 | 0.94 |
| 32 | 0.39 | 0.84 | 0.61 | 0.02 | 0.36 | 0.41 | 0.19 | 0.44 | 0.17 | 0.72 |
| 36 | 0.96 | 0.90 | 0.20 | 0.04 | 0.90 | 0.84 | 0.34 | 0.73 | 0.47 | 0.19 |
| 41 | 0.07 | 0.21 | 0.06 | 0.13 | 0.70 | 0.76 | 0.78 | 0.11 | 0.32 | 0.09 |
| 45 | 0.19 | 0.27 | 0.11 | 0.14 | 0.65 | 0.68 | 0.18 | 0.12 | 0.52 | 0.09 |
| 50 | 0.58 | 0.37 | 0.40 | 0.24 | 0.44 | 0.41 | 0.26 | 0.29 | 0.15 | 0.23 |
| 54 | 0.66 | 0.65 | 0.28 | 0.63 | 0.93 | 0.61 | 0.80 | 0.53 | 0.76 | 0.16 |
| 59 | 0.85 | 0.54 | 0.88 | 0.26 | 0.12 | 0.14 | 0.29 | 0.29 | 0.26 | 0.59 |
| 62 | 0.74 | 0.42 | 0.09 | 0.12 | 0.59 | 0.66 | 0.73 | 0.74 | 0.92 | 0.98 |
| 66 | 0.56 | 0.54 | 0.17 | 0.11 | 0.37 | 0.40 | 0.34 | 0.56 | 0.35 | 0.28 |
| 68 | 0.36 | 0.31 | 0.93 | 0.08 | 0.46 | 0.22 | | 0.45 | 0.73 | 0.66 |



**Table 4: Correlation coefficient r (Spearman) of trends for the whole core (= overall slope) and for separated sections; Degerö (DDC), Lakkasuo (ombrotrophic (LOD), minerotrophic (LMD), Breitlohmisse (Br2, Br3, Br4), Rotmeer (Ro2, Ro3), Ursee (Ur1), rewetted horizon (RW), upper mesotelm (UM), lower mesotelm (LM), catotelm (Cat)**


| site | natural overall | drained overall | RW | UM | LM | Cat |
|------|-----------------|-----------------|------|------|------|------|
| DDC | 0.35 | 0.17 | 0.98 | 0.99 | 0.95 | 0.29 |
| LOD | 0.25 | 0.07 | | 0.95 | 0.99 | 0.15 |
| LMD | 0.52 | 0.08 | 0.50 | 0.91 | 1.00 | 0.45 |
| Br2 | 0.56 | 0.29 | | 0.99 | 0.87 | 0.22 |
| Br3 | 0.56 | 0.01 | | 0.85 | 0.94 | 0.26 |
| Br4 | 0.56 | 0.23 | | 0.89 | 1.00 | 0.07 |
| Ro2 | 0.00 | 0.11 | | 0.73 | 0.50 | 0.63 |
| Ro3 | 0.11 | 0.04 | | 0.93 | 0.99 | 0.56 |
| Ur1 | 0.13 | 0.00 | 0.01 | 0.73 | 0.98 | 0.04 |
| average | 0.34 | 0.11 | 0.50 | 0.89 | 0.91 | 0.30 |





**Table 5: Trend values of all sites [‰/cm]; slopes are given for the whole core (=overall) and for separated sections (rewetted horizon (RW), upper mesotelm (UM), lower mesotelm (LM), catotelm (Cat)); Degerö (DDC), Lakkasuo (ombrotrophic (LOD), minerotrophic (LMD), Breitlohmisse (Br2, Br3, Br4), Rotmeer (Ro2, Ro3), Ursee (Ur1))**

| site | natural overall | drained overall | RW | UM | LM | Cat |
|------|------|------|------|------|------|------|
| DDC | 9.66 | 5.83 | -7.08 | 1.58 | -2.04 | 11.24 |
| LOD | 8.47 | 6.14 | | 4.06 | -5.10 | -9.49 |
| LMD | 25.18 | -6.00 | -16.95 | 3.17 | -1.78 | -10.53 |
| Br2 | 23.82 | 8.39 | | 2.19 | -2.06 | -8.80 |
| Br3 | 23.82 | -3.28 | | 4.34 | -2.25 | 6.43 |
| Br4 | 23.82 | 9.06 | | 3.22 | -3.66 | 3.34 |
| Ro2 | 3.21 | -4.58 | | 2.09 | -2.78 | -9.24 |
| Ro3 | 4.83 | -2.57 | | 0.38 | -1.89 | -11.39 |
| Ur1 | 7.76 | 0.98 | 2.52 | 13.76 | -4.82 | 2.78 |



**Table 6: Humification Indices (HI) after von Post for all investigated sites; Degerö (natural (DNM), drained (DDC)), Lakkasuo (ombrotrophic natural (LON), ombrotrophic drained (LOD), minerotrophic natural (LMN), minerotrophic drained (LMD)), Breitlohmisse (natural (Br1), drained (Br2, Br3, Br4)), Ursee (natural (Ur2), drained (Ur1)), Rotmeer (natural (Ro1), drained (Ro2, Ro3))**

| site | natural | rewetted horizon | mesotelm | catotelm |
|------|---------|------------------|----------|----------|
| DNM | H1-2 | | | H3 |
| DDC | | H2-H3 | H4 | H3 |
| LON | H2 | | | H3 |
| LOD | | | H4-H5 | H3 |
| LMN | H2 | | | H3 |
| LMD | | H3 | H4 | H3 |
| BR1 | H3 | | | H3 |
| BR2 | | | H4 | H2-H3 |
| BR3 | | | H4 | H3 |
| BR4 | | | H4 | H3 |
| UR1 | | H3 | H3-H4 | H2-H3 |
| UR2 | H2 | | | H2 |
| Ro1 | H2 | | | H3 |
| Ro2 | | | H4 | H3 |
| Ro3 | | | H4 | H3 |



**Table 7: Description of vegetation of four of the study sites; Sphagnum mosses (Sph.)**

| Site | Horizon | Main species | Description |
|---|---|---|---|
| **Degerö (DDC)** | rewetted horizon | *Sph. balticum* | Yellow, good preserved Sph.-turf, detached Sph. cymbifolia, *Vaccinium oxycoccos Eriophorum vaginatum, Andromeda polifolia* & *Cladopodiella fluitans* |
| | mesotelm | *Sph. balticum* | Darker, grayish*;* Sph.-turf, some *Eriophorum vaginatum,* detached Sph. cymbifolia, *Vaccinium oxycoccos, Andromeda polifolia* |
| | catotelm | *Sph. balticum* | Yellow; Sph.-turf, more Sph. cymbifolia, some *Eriophorum vaginatum,* detached, *Vaccinium oxycoccos, Andromeda polifolia* |
| **Lakkasuo (LOD)** | upper mesotelm | *Sph. rubellum* | Dark brown; Sph.-turf, mostly *Sph. rubellum* with *Pleutrozium schreberi* in the uppermost part |
| | lower mesotelm | *Sph. rubellum* | Dark brown, grayish; Sph.-turf, mostly *Sph. rubellum* and *Sph. balticum* |
| | catotelm | *Sph. rubellum* | Light brown, yellow; Sph.-turf, mostly *Sph. rubellum* |
| **Breitlohmisse (Br2)** | upper meostelm | *Sph. capilifolium* | Brown; Sph.-turf mostly *Sph. capilifolium* and *Sph. cymbifolia*, much *Ericaceous roots* and some *Eriophorum vaginatum* stems |
| | mesotelm | *Sph. cymbifolia* | Dark brown; Sph.-turf mostly *Sph. capilifolium* and *Sph. cymbifolia*, some *Ericaceous roots* and *Eriophorum vaginatum* |
| | catotelm | *Sph. cymbifolia* | Lighter, reddish; Sph.-turf mostly *Sph. capilifolium* and *Sph. cymbifolia*, some *Sph. acutifolia* |
| **Rotmeer (Ro2)** | upper mesotlem | *Sph. acutifolia* | Brown-reddish, yellow; Sph.-turf, mostly *Sph. acutifolia,* some *Sph. rubellum,* detached *Eriophorum vaginatum* |
| | mesotelm | *Sph. cymbifolia* | Dark brown, grayish; Sph.-turf, mostly *Sph. cymbifolia,* and *Sph. acutifolia,* some *Sph. rubellum,* detached *Eriophorum vaginatum* |
| | catotelm | *Sph. cymbifolia* | Reddish, yellow; Sph.-turf, mostly *Sph. cymbifolia,* some *Sph. acutifolia,* detached *Eriophorum vaginatum* |
