# Peer review of "Switch of fungal to bacterial degradation in natural, drained and rewetted oligotrophic peatlands reflected in $\delta^{15}N$ and fatty acid composition"

_SOIL, 2019_

## Referee Comment (RC1) · Anonymous Referee #1 · 1 Feb 2020

Groß-Schmölders and co-authors studied stable isotope profiles in intact, drained, and rewetted peatlands. They report finding a distinct maximum of d15N values present in drained but absent in natural peatlands. This maximum coincided with other factors of maximum decomposition (C:N, bulk density), as well as with the transition from fungal to bacterial dominance.

The authors present compelling evidence in support of an interesting and so far un-described phenome unique to degraded peatlands, which surely is of great interest to the SOIL readership. The mansucript is clearly written and easy to read. In my view,

however, the manuscript fails to proof many of the authors central conclusions. I think this can be addressed by the authors by slightly changing the 'angle' of the paper. I also I also have concerns with regards to the authors methodology (PLFA analysis), but I don't think these are critical to the papers key finding. Overall, I think that this paper requires a major revision before publication.

1. The authors' central hypothesis ('maximum 15N enrichment at maximum microbial diversity) seems to come out of nowhere, and it is unclear to me how the authors came up with this hypothesis except as a post-hoc justifying their results. I do not see why greater microbial diversity should necessarily imply greater nutrient limitation as it matters little to the nutrient whether it is taken up by fungi or bacteria (fungi compete with themselves as much for nutrients as with bacteria and vice versa, abundance does not equal activity, etc.). It is also unclear to me how this conclusion is supported by the presented data, which shows that the 15N maximum occurs at the same depth of the change fungal to bacterial dominance, but does not provide evidence that one is related to the other. I don't see why these changes in microbial community composition would provide evidence for greater nitrogen limitation at the depth of the 15N maximum. However, I'm not sure if this rather speculative interpretation of 15N being driven by microbial community composition is actually needed in this paper – I think describing the differences between drained and undrained peatlands provides valuable information by itself.

2. I think the manuscript could be improved by presenting/discussing the results in a different way. In the present version, the author very much focus on changes in d15N and other parameters with depth. I would recommend to start by comparing drained and undrained soils – I think it would be helpful if the authors first identify how drainage has changed the parameters measured in the soil profiles (e.g., drainage increase 15N values in the mesotelm). The authors can then discuss which processes led to an increase in 15N in drained peatlands (relative to intact neighbour sites), and why these processes were strongest in the center of the mesotelm and less pronounced towards

the surface and the catotelm edge.

3. The authors could also improve the mansucript by providing a more detailed view on the processes that cause the N isotope fractionation in these soils. In particular, they do not propose a fate for the 14N-depleted nitrogen fraction. How does this carbon get lost from the soil profile (in drained relative to intact peatlands)? It does not simply get transported downwards in the soil profile as no large difference in d15N was observed in the catotelm (Fig. 2). Mineralization is a likely mechanism, does that mean that more depleted 15N is leached out of the soil profile and exported from the peatland? Or are there stronger gaseous losses (N2O, denitrification) in drained peatlands? What is the role of plant and microbial uptake of 15N in this process?

4. PLFA analysis: The authors use a non-standard method to extract/purify/derivatize PLFAs for analysis. While this is not a problem in itself, this method looks like a total fatty acid extraction to me. At least, it extracts and recovers free fatty acids (as shown by the use of the internal standard nonadecanoic acid). Please provide information how phospholipids were separated from glycolipids and neutral lipids in this method.

Some language issues: - I would prefer the more descriptive term 'maximum' rather than 'turning point', which implies some change in direction in processes. I think this would also improves the clarity in a central point of the manuscript.

L16: 'stable isotope signatures': See this advice on 'Isotope terminology' from Z. Sharp's 'Isotope Geochemistry' book (https://digitalrepository.unm.edu/unm_oer/1/): Mistake: "The isotopic signature of the rock was d18O = 5.7‰." Recommended Expression "The d18O value of the rock was 5.7‰. Thus this rock has the oxygen isotope signature of the mantle." Explanation: "The word signature should be used to describe the isotopic composition of a significant reservoir like the mantle, the ocean, or a major part of the system being studied, not to the isotopic composition of ordinary sample" - L311: 'equilibrium' between fungi and bacteria – I don't think equilibrium is the correct concept here. Maybe change from fungal to bacterial dominance, but even that is not

necessarily true given that fungi have more biomass per unit PLFA than bacteria, and you're only looking at a very small fraction of the total PLFA pool.

Minor comments: L12-15: The first three sentences have very little to do with the content of this manuscript. L304-306: highly speculative and not well referenced. Figure 2: check axis ticks for BD and C/N, start these axis at 0. Tables 3-6 could be places in a supplement.

---

## Referee Comment (RC2) · Anonymous Referee #2 · 21 Feb 2020

General comments: The manuscript has a nicely developed and clear message. The Results and Discussion section is convincing and requires very little editing and the Figures and Tables support the message very well. To me, the Introduction is the part of the manuscript that requires the most attention. The research methods and questions need to be prepared in more detail and especially the role of roots and mycorrhiza should be considered in more detail. In summary, I rate the manuscript as GOOD and recommend publication following revisions. For consideration in the journal SOIL, the soil type/classification need to be adequately described. This is currently not the case

and needs to be done. In some figures, but also text the acrotelm-mesotelm designations are an issue: Is there an acrotelm or not? Figure 3 shows an acrotelm/upper mesotelm, but in the remainder of the manuscript, an acrotelm is not mentioned. On the other hand, an "upper" and "lower" mesotelm are introduced. Please find a consistent way to handle the issue. Please check the manuscript again for signs of sloppiness: Throughout the manuscript, the abbreviations for Tables and Figures are inconsistently spelled; sometimes capitalize and sometimes not. Somewhere in the text, I quit noting this in "Specific comments". In the references section, journal titles are generally spelled out, but sometimes not. Please edit following journal guidelines.

Specific comments: L 12: I'd rather say "Since the last centuries, they are degrading." This process is not finished. L 12: "2" in CO2 in subscript, please L 14: Why not delete "in the near future"? This is happening now. L 16: "Metabolic processes" sounds better, doesn't it? L 28: please replace "are rising" by "rise" L 32: "differs", not "differ" L 38-39: There are quite a few approaches to describe "peatland condition". But what is "peatland condition"? And are the methods you are proposing more time and cost efficient than others? You are hypothesizing that 15N isotopes could be such a tool. Fine, but PLFA analysis isn't that cheap and you are also heavily relying on that method. Please explain in more detail. L 74: "lead", not "are leading" as this is a general phenomenon Introduction chapter in general: Biogeochemical transformations as a consequence of rewetting re not introduced, but in the last paragraph of the introduction, you are looking for changes of 13C and 15N with the onset of the rewetting process. L70-82: This paragraph should be rewritten. It lists methods, but the aim/objective/hypothesis is not sufficiently clear. What are "specific peatland conditions" (Line 70)? Many methods are listed without having been introduced before. Please introduce these methods. When looking at d15N, not only decomposition must be considered, but also mycorrhizal activity. Are you expecting root effects on d15N? Chapter 2.2. Coding of the sites is inconsistent. Some codes appear to relate to minerotrophic or ombrotrophic hydrology or drained vs. natural status, but others don't. Please code in a consistent way. Chapter 2.3: What does LOD1, LON3, DDC3, DNM1 mean? Did you take replicate cores

at these sites? From which depth were samples taken? L 219: Fig., not fig. L 221; distinctly, not distinct; seem, not seems L 247: Tab, not tab (2x) L 250: "more strongly", not "more" L 283-284: This sentence is incomprehensible. Does fungal biomass decrease in peatlands? Where? When? Please explain. L 287: Drollinger, not Dröllinger L 328: "change" instead of "are changing"

References section: L 246, 420: Biogeosciences, not Biogeoscience L 443: The journal is called Wetlands Ecology and Management! L 444: Post drainage! Table 2: Sph. capillifolium, not capilifolium Supplementary data: This xls. file is not for publication. It requires formatting and translation.

---

## Author Comment (AC1) · 25 Feb 2020

Dear Referee,

Thank you, for your helpful comments, which will help to improve our paper considerably.

1.The authors' central hypothesis ('maximum 15N enrichment at maximum microbial diversity) seems to come out of nowhere, and it is unclear to me how the authors came up with this hypothesis except as a post-hoc justifying their results. I do not see why

greater microbial diversity should necessarily imply greater nutrient limitation as it matters little to the nutrient whether it is taken up by fungi or bacteria (fungi compete with themselves as much for nutrients as with bacteria and vice versa, abundance does not equal activity, etc.). It is also unclear to me how this conclusion is supported by the presented data, which shows that the 15N maximum occurs at the same depth of the change fungal to bacterial dominance, but does not provide evidence that one is related to the other. I don't see why these changes in microbial community composition would provide evidence for greater nitrogen limitation at the depth of the 15N maximum. However, I'm not sure if this rather speculative interpretation of 15N being driven by microbial community composition is actually needed in this paper – I think describing the differences between drained and undrained peatlands provides valuable information by itself.

Answer: Regarding your comment on microbial diversity and increased $\delta$15N values, we are sorry for being so unclear; we will be more precise with the description of the assumed relationship between the two parameters microbial metabolism processes and $\delta$15N values. We hypothesize that the microbial abundance (see answer 3) as well as the microbial community composition has an influence on the $\delta$15N values. With an increasing diversity, $\delta$15N values of the reaming substrate should increase, because (1) different microbial communities prefer different sources (Dijkstra et al. 2006, Drollinger et al. 2019) and (2) with increasing bacterial abundance, fungi have to use also recalcitrant sources, because bacterial metabolism will outcompete fungi for the easily degradable substances (Rousk and Bååth, 2007, Winsborough, C. and Basiliko, 2010). Hence, with increasing microbial diversity also the diversity of the mineralized organic fractions increases (Thormann 2006). With an increased diversity of nitrogen sources more release of lighter 14N is possible and the ratio of 15N:14N in the remaining substrate should increase. We realize that these are hypotheses to explain the observed patterns, but to our understanding they are the most likely ones and in describing them, we hope for an eager discussion in the community.

2. I think the manuscript could be improved by presenting/discussing the results in a different way. In the present version, the author very much focus on changes in d15N and other parameters with depth. I would recommend to start by comparing drained and undrained soils – I think it would be helpful if the authors first identify how drainage has changed the parameters measured in the soil profiles (e.g., drainage increase 15N values in the mesotelm). The authors can then discuss which processes led to an increase in 15N in drained peatlands (relative to intact neighbor sites), and why these processes were strongest in the center of the mesotelm and less pronounced towards the surface and the catotelm edge.

Answer: We have followed your advice and structured our results accordingly. We will start, as you suggested, with an overview of drained and undrained peatlands and the influence of hydrology on the measured biogeochemical parameters. This will be followed by an introduction to the processes, which leads to increased $\delta$15N values, and might explain the observed pattern in the mesotelm.

3. The authors could also improve the manuscript by providing a more detailed view on the processes that cause the N isotope fractionation in these soils. In particular, they do not propose a fate for the 14N-depleted nitrogen fraction. How does this carbon get lost from the soil profile (in drained relative to intact peatlands)? It does not simply get transported downwards in the soil profile as no large difference in d15N was observed in the catotelm (Fig. 2). Mineralization is a likely mechanism, does that mean that more depleted 15N is leached out of the soil profile and exported from the peatland? Or are there stronger gaseous losses (N2O, denitrification) in drained peatlands? What is the role of plant and microbial uptake of 15N in this process?

Answer: We for sure do not claim that we fully understand the observed patterns yet, but that we see consistent patterns and (in combination with the fatty acid analysis) develop ideas what the origin of these patterns might be. Sorry, if this was not clear from the manuscript, we will add a sentence referring to this. We will further insert a section about N isotope fractionation in peatland soils and the underlying processes,

which might be leading to 14N depletion in the remaining substrate during drainage. In general, the 15N:14N ratio of plant material (here mostly sphagnum mosses) is lower than the values of microbes and bulk material (Aldous, 2002, Lichtfouse et al. 1995). Microbes prefer to mineralize the lighter, more frequent 14N (Dijkstra et al., 2006, Novák et al, 1999). Since plants incorporate the microbial mineralized nitrogen they have a low 15N:14N ratio (Lichtfouse et al. 1995). Contrary, microbial biomass is enriched in 15N, probably as the result of processing the lighter 14N during mineralization and hence incorporation of the remaining heavier 15N. In addition, caused by the preferential mineralization of lighter nitrogen, the heavier 15N might be enriched in the remaining humic substances (Novák et al, 1999). The effect of the latter to 15N:14N bulk values is probably enhanced due to the loss of 15N-depleted material during leaching (Damman 1988, Niemen 1998), denitrification and the release of N2O (Kohzu 2003, Niemen 1998). In natural peatlands, microbial activity is low and mostly visible in the uppermost, aerobe part of the peat (acrotelm). With the onset of the waterlogging, anaerobe conditions in the catotelm microbial activity is inhibited. This leads to small or even negligible changes of the original (light) plant isotopic ratio below the acrotelm. (Dijkstra, 2008) In contrast, in drained peatlands the aerobic mesotelm expands and simultaneously microbial activity increases (Moore & Basiliko 2006, Roswell 1976). In an extended mesotelm a higher amount of mineralization and the release of N2O takes place. With increased mineralization the 15N:14N ratio in the remaining substrate should increase, as long as 14N will be mineralized preferentially (Dijkstra, 2008). However, because of the faster and more complete decomposition with increasing microbial activity (Damman, 1988) metabolism of 15N increases as well and fractionation will be less. This pattern leads to only small increases in the 15N:14N ratio of the bulk material, as all isotopes are used and fractionation is lowered in the middle of the mesotelm, where microbial activity is the highest. Actually, the best way to test for the combined effects of all these different processes on the isotopic fingerprinting would be to set up a conceptual model. However, we feel this is beyond our possibilities at the moment, but we certainly look for opportunities (e.g., cooperation)

in the future.

4. PLFA analysis: The authors use a non-standard method to extract/purify/derivatize PLFAs for analysis. While this is not a problem in itself, this method looks like a total fatty acid extraction to me. At least, it extracts and recovers free fatty acids (as shown by the use of the internal standard nonadecanoic acid). Please provide information how phospholipids were separated from glycolipids and neutral lipids in this method.

Answer: You are totally right. We have extracted all membrane fatty acids and did not separate phospholipid fatty acids. We are really sorry for this incorrect classification in the first version of our manuscript. We aimed to distinguish between fatty acids of microbes, fungi and plants and we were able to detect these changes by the extraction of total membrane fatty acid values, because the used markers (i-C15:0 and a-C-15:0 for Gram positive - bacteria and C18:2$\omega$9c for fungi) are not restricted to phospholipid fatty acids (Bajerski, Wagner & Mangelsdorf 2017; Finotti et al. 1992; Piotrowska-Seget & Mrozik 2003).

Some language issues: - I would prefer the more descriptive term 'maximum' rather than 'turning point', which implies some change in direction in processes. I think this would also improves the clarity in a central point of the manuscript.

Answer: You are right, "maximum" would also be a very good term for our observed pattern, but we decided to use "turning point" because if we compare different sites and layers, a maximum in one site or depths layer might not be the absolute maximum, which leads to confusion. Furthermore, what we are really looking at are changes in depth trends. As such, we think, turning point is a better term. Furthermore, we already used the term already to describe the observed isotope patterns in some previous publications and would thus like to stay with it.

L16: 'stable isotope signatures': See this advice on 'Isotope terminology' from Z. Sharp's 'Isotope Geochemistry' book (https://digitalrepository.unm.edu/unm_oer/1/): Mistake: "The isotopic signature of the rock was d18O = 5.7‰ ÌĞ" Recommended Expression "The d18O value of the rock was 5.7‰ÌḠhus this rock has the oxygen isotope signature of the mantle." Explanation: "The word signature should be used to describe the isotopic composition of a significant reservoir like the mantle, the ocean, or a major part of the system being studied, not to the isotopic composition of ordinary sample" - L311: 'equilibrium' between fungi and bacteria – I don't think equilibrium is the correct concept here. Maybe change from fungal to bacterial dominance, but even that is not

Answer: Thank you for the explanation for the word "signature", we have changed it to "composition". In addition, we deleted the term "equilibrium" and have decided to use "change towards higher bacterial decomposition".

Minor comments: L12-15: The first three sentences have very little to do with the content of this manuscript. L304-306: highly speculative and not well referenced. Figure 2: check axis ticks for BD and C/N, start these axis at 0. Tables 3-6 could be places in a supplement.

Answer: L12-15: We rewrote the sentences. L304-306: We added references to these sentences (Lerch et al., 2011; Rousk and Bååth, 2007). Figure 2: The axes start now with 0 and ticks are checked. Tables 3-6 will be placed in the supplement.

References

Aldous, A.R.: Nitrogen retention by Sphagnum mosses: responses to atmospheric nitrogen deposition and drought, Canadian Journal of Botany, 80, 721-731, https://doi.org/10.1139/b02-054, 2002.

Bajerski, F., Wagner, D. and Mangelsdorf, K.: Cell Membrane Fatty Acid Composition of Chryseobacterium frigidisoli PB4T. Isolated from Antarctic Glacier Forefield Soils, in Response to Changing Temperature and pH Conditions, frontiers in Microbiology, 8, 1-11, https://doi.org/10.3389/fmicb.2017.00677, 2017.

Damman, A.W.H.: Regulation of nitrogen removal and retention in Sphagnum nogs and other peatlands, OIKOS, 51, 291-305, https://www.jstor.org/stable/3565310, 1988.

Dijkstra, P., Ishizu, A., Doucett, R., Hart, S., Schwartz, E., Manyailo, O. and Hungate, B.: 13C and 15N natural abundance of the soil microbial biomass, Soil Biology & Biochemistry, 38, 3257-3266, https://doi.org/10.1016/j.soilbio.2006.04.005, 2006.

Dijkstra, P., LaViolette, C.M., Coyle, J.S., Doucett, R.R., Schwartz, E., Hart, S.C. and Hungate, B.A. (2008), 15N enrichment as an integrator of the effects of C and N on microbial metabolism and ecosystem function, Ecology Letters, 11, 389-397. https://doi.org/10.1111/j.1461-0248.2008.01154.x, 2008.

Drollinger, S., Kuzyakov, Y. and Glatzel, S.: Effects of peat decomposition on $\partial$13C and $\partial$15N depth profiles of Alpine bogs, Catena, 178, 1 – 10, https://doi.org/10.1016/j.catena.2019.02.027, 2019.

Finotti, E., Moretto, D., Marsella, R. and Mercantini: Temperature effects and fatty acid patterns in Geomyces species isolated from Antarctic soil, Polar Biology, 13, 127-130, https://doi.org/10.1007/BF00238545, 1993.

Kohzu, A., Matsui, K., Yamada, T., Sugimoto, A. and Fujita, N.: Significance of rooting depth in mire plants: Evidence from natural 15N abundance, Ecological Researxh, 18, 257–266, https://doi.org/10.1046/j.1440-1703.2003.00552.x, 2003.

Lerch, T.Z., Nunan, N, Dignac, M.-F., Chenu, C. and Mariotti, A.: Variations in microbial isotopic fractionation during soil organic matter decomposition, Biogeochemistry, 106, 5-21, DOI 10.1007/s10533-010-9432-7, 2011.

Moore, T. and Basiliko, N.: Decomposition in Boreal Peatlands, in: Boreal Peatland Ecosystems, edited by: Wieder, R.K. and Vitt, D.H., Springer, Berlin, Heidelberg, Germany, 125-143, 2006.

Niemen, M.: Changes in nitrogen cycling following the clearcutting of drained peatland forests in southern Finland, Boreal Environment, 31, 9-21, http://urn.fi/URN:NBN:fi-fe2016091423744, 1998.

Novák, M., Buzek, F. and Adamová, M.: Vertical trends in $\partial$13C, $\partial$15N and

∂34S ratios in bulk Sphagnum peat, Soil Biology and Biochemistry, 31, 1343-1346, https://doi.org/10.1016/S0038-0717(99)00040-1, 1999.

Piotrowska-Seget, Z. and Mrozik, A.: Signature Lipid Biomarker (SLB) Analysis in Determining Changes in Communitiy Structure of Soil Microorganisms, Polish Journal of Environmental Studies, 12 (6), 669-675, 2003.

Roswall, t.: The internal nitrogen cycle between microorganisms, vegetation and soil, in: Nitrogen, Phosphorous and Sulphur – Global Cycles, edited by: Svensson, B.H. and Söderlund, R., Ecology Bulleting, Stockholm, Sweden, 157-167, 1976.

Rousk, J. and Bååth, E.: Fungal biomass production and turnover in soil estimated using the acetate-in-ergosterol technique, Soil Biology & Biochemistry, 39, 2173-2177, doi:10.1016/j.soilbio.2007.03.023, 2007

Winsborough, C. and Basiliko, N.: Fungal and Bacterial Activity in Nothern Peatlands, Geochemistry Journal, 27, 315-320, https://doi.org/10.1080/01490450903424432, 2010.

—————————————————————————

---

## Author Comment (AC2) · 13 Mar 2020

Dear Referee,

Thank you, for your helpful comments, which will improve our paper considerably.

1. To me, the Introduction is the part of the manuscript that requires the most attention. The research methods and questions need to be prepared in more detail and especially the role of roots and mycorrhiza should be considered in more detail.

[Figure]

Answer: We have rewritten the introduction and have insert a more detailed part for research methods and questions (see also answer 7). You are right; we have not sufficiently discussed the expected role of roots and mycorrhiza. We are sorry for that and have now complemented a paragraph with it. Our study sites are open peatlands with a small amount of vascular plants (result of our vegetation analysis). Hence, mycorrhiza should also play a minor role. But of course, there might still be an effect of mycorrhizal activity and rooting in our study sites. Mycorrhiza mediate the uptake of nitrogen into plants. Rooting and the existence of mycorrhiza leads to enriched 15N values in the remaining bulk material (Högberg et al., 1996), because (1) mycorrhiza preferentially process lighter 14N and transfer them to plants (Adams and Grierson, 2001, Asada et al., 2005a, Högberg et al., 1996, Kohzu et al., 2003, Robinson et al., 1998) and (2), even without mycorrhizal activity, plants preferentially incorporate the lighter 14N. But, because of the small amount of vascular plants and therefore also of mycorrhiza, these cannot not be the main drivers for our observed pattern.

2. For consideration in the journal SOIL, the soil type/classification need to be adequately described.

Answer: You are right, we have forgotten to describe the soil type adequately and have inserted this classification to the manuscript. All investigated sites are classified as Histosols (organic soils). Histosols are classified as soils with a cumulative organic layer and an organic matter amount of 35% or higher in at least half of the uppermost 80-100 cm (IUSS, 2015). In addition all investigated peatland soils are Sphagnum peat, because of their mean annual temperatures (between +1.2°C and +7 °C) and their annual precipitation between 523ppm and 1600ppm (Eurola et al., 1984, Vitt et al., 2006). Lakkasuo and Degerö Stormyr are classified as Northern eccentric bogs and the peatlands in the black forest are characterized as ombrotrophic bogs (Eurola et al., 1984).

3. In some figures, but also text the acrotelm-mesotelm designations are an issue: Is there an acrotelm or not? Figure 3 shows an acrotelm/upper mesotelm, but in the

remainder of the manuscript, an acrotelm is not mentioned. On the other hand, an "upper" and "lower" mesotelm are introduced. Please find a consistent way to handle the issue.

Answer: We are sorry, for not being clear enough with our definition. Yes, there is an acrotelm and a mesotelm in all sites. We have forgotten to mention this in figure 3 and have changed it accordingly. The acrotelm is the uppermost part of the peatland with living sphagnum vegetation (Morris et al., 2011). The deeper part, with dead plant material and in permanent waterlogged, anaerobe conditions, is called catotelm (Morris et al., 2011). In between the acrotelm and the catotelm, with fluctuating conditions, the mesotelm is located (Clymo and Bryant, 2008, Lin et al., 2014). With drainage, the mesotelm is expanding and a supplementary separation is reasonable, because the condition within the mesotelm differ a lot from aerobe, light and warm conditions (upper mesotelm) to semi-oxic, dark and cold conditions in the lower mesotelm (Artz, 2014, Lin et al., 2014). These changed conditions are the reason for the changed microbial metabolic pathways and are therefore critical for the 14N:15N ratio we see in the data sets (Lin et al., 2014).

4. Please check the manuscript again for signs of sloppiness: Throughout the manuscript, the abbreviations for Tables and Figures are inconsistently spelled; sometimes capitalize and sometimes not. Somewhere in the text, I quit nothing this in "Specific comments". In the references section, journal titles are generally spelled out, but sometimes not. Please edit following journal guidelines

Answer: Thank you for pointing this out. We have checked and deleted mistakes in tables and figures as well as in the reference section.

5. Specific comments: L12; L14; L16; L28; L32; L74; L219; L221; L247; L250; L287; L246; L443; L444

Answer: Thank you for your careful and constructive review of the manuscript. We have changed and improved the mentioned sentences.

6. L38-39: There are quite a few approaches to describe "peatland condition". But what is "peatland condition"? And are the methods you are proposing more time and cost efficient than others? You are hypothesizing that 15N isotopes could be such a tool. Fine, but PLFA analysis isn't that cheap and you are also heavily relying on that method. Please explain in more detail.

Answer: With the wording "peatland conditions" we are referring to the hydrology status, whether it is natural, drained or rewetted. We have changed the wording to hydrology status. You are right; FA analysis is not a cheap and easy method. We have done this analysis to support our hypothesis based on stable isotopes, and only the latter we refer to as a time- and cost efficient method to indicate drainage and rewetting. We do not suggest establishing an approach as a routine analysis, which uses both methods. The three main methods today to measure the hydrology status are (1) a macro analysis of peatland vegetation, (2) gas emission measurements and (3) measurement of growth heights of peatland vegetation. Method (1) was also done in this study. We wanted to prove, that our investigated stable isotope patterns are related to decomposition and that they are not primarily a consequence of the vegetation assemblages. But this method is time consuming and needs a high level of expert knowledge and is thus very costly. Method (2), the measurement of gas exchange in peatlands (Baldocchi et al., 1988) measures current gas emissions and therefore provides an indirect measurement of ongoing decomposition processes (Bubier et al., 2003). But it is not able to give information on drainage history and gas exchanges at another time of the year (Bubier et al., 2003). Furthermore, this method is also very intensive in analytical equipment and expert knowledge needed. A third available method (3) is the measurement of the growth of peatland vegetation. But there are several problems with this method: Firstly, not only the sole growth of mosses indicates peat growth. It is important how much vegetation material enters the catotelm and is therefore stored under aerobe conditions. Secondly, peat shrinks and swells with water supply. Hence measuring peat height at different times would lead to completely different assumptions for peatland growth (Clymo, 1970). And thirdly, peat growth is really slow and it would

need decades to get a positive reply with this method to indicate successful restoration efforts (Clymo, 1970, Fenton, 1980). Summing up, there are methods available to get information of the success of restoration effort, but these methods are lacking some important information or/ and are expensive and time consuming. Hence, there is a need for a new and less expensive and time-consuming indicators, which could be done not only by specific experts. We believe that bulk isotopes can be such suitable indicators, but we need to prove that with the FA method.

7. This is a general phenomenon Introduction chapter in general: Biogeochemical transformations as a consequence of rewetting re not introduced, but in the last paragraph of the introduction, you are looking for changes of 13C and 15N with the onset of the rewetting process.

Answer: Thank you, for your comment. You are right; we have missed to introduce our hypothesis of the influence of rewetting to stable isotopes. Rewetting increases the water table height and therefore enlarges the anaerobe catotelm (Andersen et al., 2006). We hypothesize, that the observed stable isotope pattern for drained horizons will be conserved, when formerly aerobe parts will get rewetted (Andersen et al., 2006). With rewetting the conditions in the former mesotelm will get anaerobe and microbial activity will be inhibited (Andersen et al., 2006, Asada et al., 2005b, Thormann et al., 1999). Hence, no or only few metabolism processes take place and stable isotope patterns shouldn't change anymore. For the upper part of the rewetted peat, we expect to find natural conditions and vegetation growth, like in natural peatlands. Hence, we expect to find the same stable isotope pattern, as we see in natural peat.

8. L70-80: This paragraph should be rewritten. It lists methods, but the aim/objective/hypothesis is not sufficiently clear. Many methods are listed without having been introduced before. Please introduce these methods. When looking at d15N, not only decomposition must be considered, but also mycorrhizal activity. Are you expecting root effects on d15N?

Answer: We have improved the wording of the mentioned paragraph and inserted an introduction to the mentioned methods (bulk density and carbon:nitrogen ratio measurements). Our aim is to find an answer for the depth trends of carbon and nitrogen stable isotopes corresponding to the hydrology status, which were investigated in previous studies. Our main hypothesis is that microbial metabolic pathways are the drivers behind these stable isotope depth trends. The hydrology status determines the abundance of microbial communities. With changing hydrology microbial abundance changes significantly (Kohl et al., 2013) and therefore also stable isotope values must change (Tfaily et al., 2014). Vice versa this would mean that stable isotope values reflect the hydrology status, which we aim to test. We hypothesize, that drained conditions lead to expanded microbial abundance, because of the attendance of oxygen also in deeper horizons. We aim to find significant links between this pattern and the observed stable isotope pattern. For natural and rewetted conditions we hypothesize to find low values of stable isotopes in accordance to low microbial abundance. For mycorrhizal- and rooting effects please see answer 1.

9. Chapter 2.2. Coding of the sites is inconsistent. Some codes appear to relate to minerotrophic or ombrotrophic hydrology or drained vs. natural status, but others don't. Please code in a consistent way. Chapter 2.3: What does LOD1, LON3, DDC3, DNM1 mean? Did you take replicate cores at these sites? From which depth were samples taken?

Answer: We have changed the coding, to be more consistent. We have now named them as follows: first letter of the site + hydrology status (plus with subscript, if needed, for additional information) (Tab.1). Yes, we had three replicates per site and analyzed samples of the upper 60 cm of the cores. The cores were sliced in 2 cm sections and every second layer was analyzed, giving a 4 cm depth resolution. We have mentioned it in section 2.2 (L134/135 and L148).

10. L 283-284: This sentence is incomprehensible. Does fungal biomass decrease in peatlands? Where? When? Please explain.

Answer: Yes, our hypothesis is, that with increasing depth and changing hydrological conditions (darker, less oxygen) fungi will be outcompeted by bacteria, which means, that fungal biomass must decrease, whereas bacterial biomass increases with depth. In the uppermost part (acrotelm and upper mesotelm) fungal biomass is the highest, whereas in the deeper part of the mesotelm bacterial biomass will increase. In the catotelm all microbial biomass is strongly reduced because of the anaerobe conditions. In natural peatlands a small amount of fungal biomass is also visible in the acrotelm, but in a much lower scale than for drained sites. (Thormann, 1999)

11. Supplementary data: This xls. file is not for publication. It requires formatting and translation.

Answer: We have re-formatted the supplementary data to make it more comprehensible for readers.

References

[revised manuscript text omitted]

---

## Author Response (AR1)

Dear Referee,

Thank you, for your helpful comments, which will help to improve our paper considerably.

*1.The authors' central hypothesis ('maximum 15N enrichment at maximum microbial diversity) seems to come out of nowhere, and it is unclear to me how the authors came up with this hypothesis except as a post-hoc justifying their results. I do not see why greater microbial diversity should necessarily imply greater nutrient limitation as it matters little to the nutrient whether it is taken up by fungi or bacteria (fungi compete with themselves as much for nutrients as with bacteria and vice versa, abundance does not equal activity, etc.). It is also unclear to me how this conclusion is supported by the presented data, which shows that the 15N maximum occurs at the same depth of the change fungal to bacterial dominance, but does not provide evidence that one is related to the other. I don't see why these changes in microbial community composition would provide evidence for greater nitrogen limitation at the depth of the 15N maximum. However, I'm not sure if this rather speculative interpretation of 15N being driven by microbial community composition is actually needed in this paper – I think de- scribing the differences between drained and undrained peatlands provides valuable information by itself.*

Answer:

**Influence of microbial diversity:** Regarding your comment on microbial diversity and increased $\delta^{15}N$ values, we are sorry for being so unclear; we will be more precise with the description of the assumed relationship between the two parameters microbial metabolism processes and $\delta^{15}N$ values. We hypothesize that the microbial abundance (see answer 3) as well as the microbial community composition has an influence on the $\delta^{15}N$ values. With an increasing diversity, $\delta^{15}N$ values of the reaming substrate should increase, because (1) different microbial communities prefer different sources (Dijkstra et al. 2006, Dröllinger et al. 2019) and (2) with increasing bacterial abundance, fungi have to use also recalcitrant sources, because bacterial metabolism will outcompete fungi for the easily degradable substances (Rousk and Bååth, 2007, Winsborough, C. and Basiliko, 2010). Hence, with increasing microbial diversity also the diversity of the mineralized organic fractions increases (Thormann 2006). With an increased diversity of nitrogen sources more release of lighter $^{14}N$ is possible and the ratio of $^{15}N:^{14}N$ in the remaining substrate should increase. We realize that these are hypotheses to explain the observed patterns, but to our understanding they are the most likely ones and in describing them, we hope for an eager discussion in the community.

We have implemented this to the manuscript in section 3.6 (L501-508).

*2. I think the manuscript could be improved by presenting/discussing the results in a different way. In the present version, the author very much focus on changes in d15N and other parameters with depth. I would recommend to start by comparing drained and undrained soils – I think it would be helpful if the authors first identify how drainage has changed the parameters measured in the soil profiles (e.g., drainage increase 15N values in the mesotelm). The authors can then discuss which processes led to an increase in 15N in drained peatlands (relative to intact neighbor sites), and*

Answer:
**Different way to present:** We have followed your advice and structured our results accordingly. We will start, as you suggested, with an overview of drained and undrained peatlands and the influence of hydrology on the measured biogeochemical parameters. This will be followed by an introduction to the processes, which leads to increased $\delta^{15}N$ values, and might explain the observed pattern in the mesotelm.

We have implemented these changes and restructured chapter 3 accordingly.

*3. The authors could also improve the manuscript by providing a more detailed view on the processes that cause the N isotope fractionation in these soils. In particular, they do not propose a fate for the 14N-depleted nitrogen fraction. How does this carbon get lost from the soil profile (in drained relative to intact peatlands)? It does not simply get transported downwards in the soil profile as no large difference in d15N was observed in the catotelm (Fig. 2). Mineralization is a likely mechanism, does that mean that more depleted 15N is leached out of the soil profile and exported from the peatland? Or are there stronger gaseous losses (N2O, denitrification) in drained peatlands? What is the role of plant and microbial uptake of 15N in this process?*

Answer:
**Nitrogen cycling:** We for sure do not claim that we fully understand the observed patterns yet, but that we see consistent patterns and (in combination with the fatty acid analysis) develop ideas what the origin of these patterns might be. Sorry, if this was not clear from the manuscript, we will add a sentence referring to this. We will further insert a section about N isotope fractionation in peatland soils and the underlying processes, which might be leading to $^{14}N$ depletion in the remaining substrate during drainage. In general, the $^{15}N:^{14}N$ ratio of plant material (here mostly sphagnum mosses) is lower than the values of microbes and bulk material (Aldous, 2002, Lichtfouse et al. 1995). Microbes prefer to mineralize the lighter, more frequent $^{14}N$ (Dijkstra et al., 2006, Novák et al, 1999). Since plants incorporate the microbial mineralized nitrogen they have a low $^{15}N:^{14}N$ ratio (Lichtfouse et al. 1995). Contrary, microbial biomass is enriched in $^{15}N$, probably as the result of processing the lighter $^{14}N$ during mineralization and hence incorporation of the remaining heavier $^{15}N$. In addition, caused by the preferential mineralization of lighter nitrogen, the heavier $^{15}N$ might be enriched in the remaining humic substances (Novák et al, 1999). The effect of the latter to $^{15}N:^{14}N$ bulk values is probably enhanced due to the loss of $^{15}N$-depleted material during leaching (Damman 1988, Niemen 1998), denitrification and the release of $N_2O$ (Kohzu 2003, Niemen 1998).
In natural peatlands, microbial activity is low and mostly visible in the uppermost, aerobe part of the peat (acrotelm). With the onset of the waterlogging, anaerobe conditions in the catotelm microbial activity is inhibited. This leads to small or even negligible changes of the original (light) plant isotopic ratio below the acrotelm. (Dijkstra, 2008)
In contrast, in drained peatlands the aerobic mesotelm expands and simultaneously microbial activity increases (Moore & Basiliko 2006, Roswell 1976). In an extended mesotelm a higher amount of mineralization and the release of $N_2O$ takes place. With

increased mineralization the $^{15}N:^{14}N$ ratio in the remaining substrate should increase, as long as $^{14}N$ will be mineralized preferentially (Dijkstra, 2008). However, because of the faster and more complete decomposition with increasing microbial activity (Damman, 1988) metabolism of $^{15}N$ increases as well and fractionation will be less. This pattern leads to only small increases in the $^{15}N:^{14}N$ ratio of the bulk material, as all isotopes are used and fractionation is lowered in the middle of the mesotelm, where microbial activity is the highest. Actually, the best way to test for the combined effects of all these different processes on the isotopic fingerprinting would be to set up a conceptual model. However, we feel this is beyond our possibilities at the moment, but we certainly look for opportunities (e.g., cooperation) in the future.

We have implemented this mainly in chapter 3.6 (L463 – 514).

*4. PLFA analysis: The authors use a non-standard method to extract/purify/derivatize PLFAs for analysis. While this is not a problem in itself, this method looks like a total fatty acid extraction to me. At least, it extracts and recovers free fatty acids (as shown by the use of the internal standard nonadecanoic acid). Please provide information how phospholipids were separated from glycolipids and neutral lipids in this method.*

Answer:
**FA analysis:** You are totally right. We have extracted all membrane fatty acids and did not separate phospholipid fatty acids. We are really sorry for this incorrect classification in the first version of our manuscript. We aimed to distinguish between fatty acids of microbes, fungi and plants and we were able to detect these changes by the extraction of total membrane fatty acid values, because the used markers (i-C15:0 and a-C-15:0 for Gram positive - bacteria and C18:2ω9c for fungi) are not restricted to phospholipid fatty acids (Bajerski, Wagner & Mangelsdorf 2017; Finotti et al. 1992; Piotrowska-Seget & Mrozik 2003).

We have implemented this in fatty acid sections 2.3, 3.5 and for the whole manuscript, especially for L 27-36 and L128-138.

*Some language issues: - I would prefer the more descriptive term 'maximum' rather than 'turning point', which implies some change in direction in processes. I think this would also improves the clarity in a central point of the manuscript.*

Answer:
**Language issue:** You are right, "maximum" would also be a very good term for our observed pattern, but we decided to use "turning point" because if we compare different sites and layers, a maximum in one site or depths layer might not be the absolute maximum, which leads to confusion. Furthermore, what we are really looking at are changes in depth trends. As such, we think, turning point is a better term. Furthermore, we already used the term already to describe the observed isotope patterns in some previous publications and would thus like to stay with it.

*L16: 'stable isotope signatures': See this advice on 'Isotope terminology' from Z. Sharp's 'Isotope Geochemistry' book (https://digitalrepository.unm.edu/unm_oer/1/): Mistake: "The isotopic signature of the rock was d18O = 5.7‰" Recommended Expression "The d18O value of the rock was 5.7‰ Thus this rock has the oxygen isotope signature of the mantle." Explanation: "The word signature should be used to*

*describe the isotopic composition of a significant reservoir like the mantle, the ocean, or a major part of the system being studied, not to the isotopic composition of ordinary sample" - L311: 'equilibrium' between fungi and bacteria – I don't think equilibrium is the correct concept here. Maybe change from fungal to bacterial dominance, but even that is not*

Answer:
Thank you for the explanation for the word "signature", we have changed it to "composition". In addition, we deleted the term "equilibrium" and have decided to use "change towards higher bacterial decomposition".

*Minor comments: L12-15: The first three sentences have very little to do with the content of this manuscript. L304-306: highly speculative and not well referenced. Figure 2: check axis ticks for BD and C/N, start these axis at 0. Tables 3-6 could be places in a supplement.*

Answer:
L12-15: We rewrote the sentences.
L304-306: We added references to these sentences (Lerch et al., 2011; Rousk and Bååth, 2007).
Figure 2: The axes start now with 0 and ticks are checked.
Tables 3-6 will be placed in the supplement.

Answer: We have changed the coding, to be more consistent. We have now named them as follows: first letter of the site + hydrology status (plus with subscript, if needed, for additional information) (Tab.1). Yes, we had three replicates per site and analyzed samples of the upper 60 cm of the cores. The cores were sliced in 2 cm sections and every second layer was analyzed, giving a 4 cm depth resolution. We have mentioned it in section 2.2 (L134/135 and L148).

**Table 1: Labeling of all drilling sites**

| Location | Labeling |
|---|---|
| Degerö | |
| natural mire | DN |
| drained | DD |
| Lakkasuo | |
| minerotrophic natural | $LN_m$ |
| minerotrophic drained | $LD_m$ |
| ombrotrophic natural | $LN_o$ |
| ombrotrophic drained | $LD_o$ |
| Breitlohmisse | |
| natural mire | BN |

| | |
|---|---|
| natural dry | $BN_{dry}$ |
| drained | $BD_1$ |
| near the mire edge | $BD_2$ |

| | |
|---|---|
| Rotmeer | |
| natural mire | RN |
| drained, with Sphagnum | $RD_1$ |
| drained, without Sphagnum | $RD_2$ |

| | |
|---|---|
| Ursee | |
| natural mire | UN |
| drained | UD |

*10. L 283-284: This sentence is incomprehensible. Does fungal biomass decrease in peatlands? Where? When? Please explain.*

Answer: Yes, our hypothesis is, that with increasing depth and changing hydrological conditions (darker, less oxygen) fungi will be outcompeted by bacteria, which means, that fungal biomass must decrease, whereas bacterial biomass increases with depth. In the uppermost part (acrotelm and upper mesotelm) fungal biomass is the highest, whereas in the deeper part of the mesotelm bacterial biomass will increase. In the catotelm all microbial biomass is strongly reduced because of the anaerobe conditions. In natural peatlands a small amount of fungal biomass is also visible in the acrotelm, but in a much lower scale than for drained sites. (Thormann, 1999)

We have implemented it in the chapters 3.5 (L440-457) and 3.6 (L479-491).

*11. Supplementary data: This xls. file is not for publication. It requires formatting and translation.*

Answer: We have re-formatted the supplementary data to make it more comprehensible for readers.

Answer:

**Nitrogen cycling:** We for sure do not claim that we fully understand the observed patterns yet, but that we see consistent patterns and (in combination with the fatty acid analysis) develop ideas what the origin of these patterns might be. Sorry, if this was not clear from the manuscript, we will add a sentence referring to this. We will further insert a section about N isotope fractionation in peatland soils and the underlying processes, which might be leading to $^{14}N$ depletion in the remaining substrate during drainage. In general, the $^{15}N{:}^{14}N$ ratio of plant material (here mostly sphagnum mosses) is lower than the values of microbes and bulk material (Aldous, 2002, Lichtfouse et al. 1995). Microbes prefer to mineralize the lighter, more frequent $^{14}N$ (Dijkstra et al., 2006, Novák et al, 1999). Since plants incorporate the microbial mineralized nitrogen they have a low $^{15}N{:}^{14}N$ ratio (Lichtfouse et al. 1995). Contrary, microbial biomass is enriched in $^{15}N$, probably as the result of processing the lighter $^{14}N$ during mineralization and hence incorporation of the remaining heavier $^{15}N$. In addition, caused by the preferential mineralization of lighter nitrogen, the heavier $^{15}N$ might be enriched in the remaining humic substances (Novák et al, 1999). The effect of the latter to $^{15}N{:}^{14}N$ bulk values is probably enhanced due to the loss of $^{15}N$-depleted material during leaching (Damman 1988, Niemen 1998), denitrification and the release of $N_2O$ (Kohzu 2003, Niemen 1998).

In natural peatlands, microbial activity is low and mostly visible in the uppermost, aerobe part of the peat (acrotelm). With the onset of the waterlogging, anaerobe conditions in the catotelm microbial activity is inhibited. This leads to small or even negligible changes of the original (light) plant isotopic ratio below the acrotelm. (Dijkstra, 2008)

In contrast, in drained peatlands the aerobic mesotelm expands and simultaneously microbial activity increases (Moore & Basiliko 2006, Roswell 1976). In an extended mesotelm a higher amount of mineralization and the release of $N_2O$ takes place. With

increased mineralization the $^{15}$N:$^{14}$N ratio in the remaining substrate should increase, as long as $^{14}$N will be mineralized preferentially (Dijkstra, 2008). However, because of the faster and more complete decomposition with increasing microbial activity (Damman, 1988) metabolism of $^{15}$N increases as well and fractionation will be less. This pattern leads to only small increases in the $^{15}$N:$^{14}$N ratio of the bulk material, as all isotopes are used and fractionation is lowered in the middle of the mesotelm, where microbial activity is the highest. Actually, the best way to test for the combined effects of all these different processes on the isotopic fingerprinting would be to set up a conceptual model. However, we feel this is beyond our possibilities at the moment, but we certainly look for opportunities (e.g., cooperation) in the future.

We have implemented this mainly in chapter 3.6 (L463 – 514).

*4. PLFA analysis: The authors use a non-standard method to extract/purify/derivatize PLFAs for analysis. While this is not a problem in itself, this method looks like a total fatty acid extraction to me. At least, it extracts and recovers free fatty acids (as shown by the use of the internal standard nonadecanoic acid). Please provide information how phospholipids were separated from glycolipids and neutral lipids in this method.*

Answer:
**FA analysis:** You are totally right. We have extracted all membrane fatty acids and did not separate phospholipid fatty acids. We are really sorry for this incorrect classification in the first version of our manuscript. We aimed to distinguish between fatty acids of microbes, fungi and plants and we were able to detect these changes by the extraction of total membrane fatty acid values, because the used markers (i-C15:0 and a-C-15:0 for Gram positive - bacteria and C18:2ω9c for fungi) are not restricted to phospholipid fatty acids (Bajerski, Wagner & Mangelsdorf 2017; Finotti et al. 1992; Piotrowska-Seget & Mrozik 2003).

We have implemented this in fatty acid sections 2.3, 3.5 and for the whole manuscript, especially for L 27-36 and L128-138.

*Some language issues: - I would prefer the more descriptive term 'maximum' rather than 'turning point', which implies some change in direction in processes. I think this would also improves the clarity in a central point of the manuscript.*

Answer:
**Language issue:** You are right, "maximum" would also be a very good term for our observed pattern, but we decided to use "turning point" because if we compare different sites and layers, a maximum in one site or depths layer might not be the absolute maximum, which leads to confusion. Furthermore, what we are really looking at are changes in depth trends. As such, we think, turning point is a better term. Furthermore, we already used the term already to describe the observed isotope patterns in some previous publications and would thus like to stay with it.

*L16: 'stable isotope signatures': See this advice on 'Isotope terminology' from Z. Sharp's 'Isotope Geochemistry' book (https://digitalrepository.unm.edu/unm_oer/1/): Mistake: "The isotopic signature of the rock was d18O = 5.7‰" Recommended Expression "Thed18Ovalueoftherockwas5.7‰T husthisrockhastheoxygenisotope signature of the mantle." Explanation: "The word signature should be used to*

*describe the isotopic composition of a significant reservoir like the mantle, the ocean, or a major part of the system being studied, not to the isotopic composition of ordinary sample"* - *L311: 'equilibrium' between fungi and bacteria – I don't think equilibrium is the correct concept here. Maybe change from fungal to bacterial dominance, but even that is not*

Answer:
Thank you for the explanation for the word "signature", we have changed it to "composition". In addition, we deleted the term "equilibrium" and have decided to use "change towards higher bacterial decomposition".

*Minor comments: L12-15: The first three sentences have very little to do with the content of this manuscript. L304-306: highly speculative and not well referenced. Figure 2: check axis ticks for BD and C/N, start these axis at 0. Tables 3-6 could be places in a supplement.*

Answer:
L12-15: We rewrote the sentences.
L304-306: We added references to these sentences (Lerch et al., 2011; Rousk and Bååth, 2007).
Figure 2: The axes start now with 0 and ticks are checked.
Tables 3-6 will be placed in the supplement.

Answer: We have changed the coding, to be more consistent. We have now named them as follows: first letter of the site + hydrology status (plus with subscript, if needed, for additional information) (Tab.1). Yes, we had three replicates per site and analyzed samples of the upper 60 cm of the cores. The cores were sliced in 2 cm sections and every second layer was analyzed, giving a 4 cm depth resolution. We have mentioned it in section 2.2 (L134/135 and L148).

**Table 1: Labeling of all drilling sites**

| Location | Labeling |
|---|---|
| Degerö | |
| natural mire | DN |
| drained | DD |
| Lakkasuo | |
| minerotrophic natural | $LN_m$ |
| minerotrophic drained | $LD_m$ |
| ombrotrophic natural | $LN_o$ |
| ombrotrophic drained | $LD_o$ |
| Breitlohmisse | |
| natural mire | BN |

| | |
|---|---|
| natural dry | $BN_{dry}$ |
| drained | $BD_1$ |
| near the mire edge | $BD_2$ |
| Rotmeer | |
| natural mire | RN |
| drained, with Sphagnum | $RD_1$ |
| drained, without Sphagnum | $RD_2$ |
| Ursee | |
| natural mire | UN |
| drained | UD |

*10. L 283-284: This sentence is incomprehensible. Does fungal biomass decrease in peatlands? Where? When? Please explain.*

Answer: Yes, our hypothesis is, that with increasing depth and changing hydrological conditions (darker, less oxygen) fungi will be outcompeted by bacteria, which means, that fungal biomass must decrease, whereas bacterial biomass increases with depth. In the uppermost part (acrotelm and upper mesotelm) fungal biomass is the highest, whereas in the deeper part of the mesotelm bacterial biomass will increase. In the catotelm all microbial biomass is strongly reduced because of the anaerobe conditions. In natural peatlands a small amount of fungal biomass is also visible in the acrotelm, but in a much lower scale than for drained sites. (Thormann, 1999)

We have implemented it in the chapters 3.5 (L440-457) and 3.6 (L479-491).

*11. Supplementary data: This xls. file is not for publication. It requires formatting and translation.*

Answer: We have re-formatted the supplementary data to make it more comprehensible for readers.

[revised manuscript text omitted]

---

## Author Response (AR2)

Dear Dr. Jeanette Whitaker,

Thank you very much. We carefully considered all comments and suggestions. Please find a detailed list with answers and comments on the revisions from Reviewer 1 below. (Line numbers are referring to the document with track changes)

L 39-40: The best tools to describe peatland hydrology would be water table loggers. I think that you don't really want to say that isotopic signatures are the preferred way to provide information on hydrology.

Our analysis of the hydrology status is not referring to the current water table height, but we target to describe the status and metabolism of the peat soil under drained, rewetted and natural conditions. Sorry if this was not clear, we have defined this now in Lines 41-42 to avoid misunderstandings.

L 41: The designation of peat soil horizons may be approached from very different perspectives. The acrotelm-mesotelm-catotelm concept is one of them. Thus, I suggest dampening that sentence to something like "From a hydrologic perspective, peatland soils maybe divided in three horizons."

Thank you. We have rewritten the sentence as you suggested (L43).

L 59-61: Here, you are implying that macrofossil analyses would be a preferred method, but for what? For determining humification?

Thank you for your comment. We specified the sentence (L: 59-60). We referred to this analysis as a tool to get information on the peatland hydrological status and the degree of decomposition. In the past, we several times got the comment to determine vegetation/ macrofossil analysis instead of measuring isotope values. In fact, it is easier and cheaper to measure isotopes than finding experts familiar with peat vegetation/ macrofossil analysis.

L 62-73: The purpose of gas emission measurements and measurements of growth heights of peatland vegetation are not to measure peatland hydrology. I'd say that water table depth, soil moisture, the energy/water budget and some other methods serve that purpose. Gas emission measurements and growth heights of peatland vegetation are proxies for some parameters as peatland hydrology, as are other types of data like vegetation species and their indicator for site water supply (eg GEST concept) and others. I don't see the purpose of this paragraph.

Thank you for your suggestions. We were actually asked to insert this section by the second Reviewer, to show alternative methods to get information on the hydrology status. Please note, that we refer again to the term hydrological status in terms of metabolic state, water and oxygen saturation state. It is important to name them as alternative measurement possibilities to get information for ongoing or recent decomposition processes, which are correlated with the hydrology status, as we define it.

L 74: Are you still referring to indicators for hydrology here?

Thank you for your comment. Yes, we do and have specified the sentence (L 75-76).

L 80: "drained, rewetted and natural" are not sharp categories describing peatland hydrology. Rewetting and drainage (how deep?) may result in very different hydrologic situations, and natural conditions may be very variable in a hydrologic sense as well.

Thank you for your comment. We have separated the categories, as mentioned in the lines 57-59 and section to, with the help of available data (water table height, vegetation analysis) and with historical information of the installation of drainage channels. We inserted this information also now in line 59.

L 99-100: Open peatlands often have a lot of vascular plants. I know Lakkasuo and Degerö Stormy, there is a lot of shrubs there. Or did you sample from a Sphagnum lawn region there? Even then, peat at some depth may be influenced by shrubs having grown there before today. You also show in Tabkle 3 that shrubs are common. I think it is quite speculative to assume that mycorrhiza do not play a role here. Maybe you should say that you "assume that these mechanisms cannot be the main drivers".

We concentrated on Sphagnum lawn regions with our sampling. But you are right, the suggested formulation describes the situation better and we changed it (L 104).

L 125/126: Please give soil classification used (WRB?) and add a qualifier to the reference soil group (Fibric/Sapric?) This would be helpful for the discussion on decomposition.

We have inserted it in the lines 129 - 132. The peatland soils are classified as fibric Histosols (HSf).

L 310: Replace by "natural" by "oligotrophic". Many natural peatlands a C/N ratio of <50.

Thank you for the comment. We have replaced it. (L: 314)

L 371: Delete "were"

We deleted it. (L: 375)

L 381: "metabolic" instead of "metabolism"

It is changed. (L: 385)

L 390: It would be good so see a graph of this correlation. 0.4 r2 is not necessarily a very good correlation.

Thank you for your comment. We have inserted a graph (Figure 3)

L 419: "stated", not "stated out"

It is changed. (L: 424)

L 430-432: I can't follow this reasoning here. Why should the diversity of mineralize organic fractions increase? Thormann et al., 2006 examined yeasts.

We have inserted the wrong date. The publication is from 2005 not 2006 (Thormann, M.: Diversity and function of fungi in peatlands: A carbon cycling perspective, Canadian Journal of Soil Science, 281-293, https://doi.org/10.4141/S05-082, 2005.) Sorry for this, we corrected it. (L 436)

L 459: Reconsider the tense used. Why "are changing"?

It is right, "were changed" might be a better tense. It is changed. (L: 464)

[revised manuscript text omitted]